# REPRESENTATION CONSOLIDATION FROM MULTIPLE EXPERT TEACHERS

## ABSTRACT

A library of diverse expert models transfers better to a novel task than a single generalist model. However, growing such a library indefinitely is impractical. Hence, we explore the problem of learning a consolidated image feature representation from a collection of related task-specific teachers that transfer well on novel recognition tasks. This differs from traditional knowledge distillation in which a student model is trained to emulate the input/output functionality of a teacher. Indeed, we observe experimentally that standard distillation of task-specific teachers, or using these teacher representations directly, *reduces* downstream transferability compared to a task-agnostic generalist model. We show that a simple multi-head, multi-task distillation method using an unlabeled proxy dataset and adding a generalist teacher is sufficient to consolidate representations from task-specific teacher(s). We improve downstream performance, outperforming the teacher (or best of all teachers) as well as the strong baseline of ImageNet pre-trained features. Our method almost reaches the performance of a multi-task joint training oracle, reaping the benefit of the teachers without replaying their training data.

## 1 INTRODUCTION

A promising approach to scale transfer learning to diverse downstream vision tasks is to maintain a library of diverse experts pre-trained on different tasks. (Deshpande et al., 2021) When presented with a novel downstream task, one can select an appropriate expert and fine-tune the representation with small amounts of task-specific data. This strategy has many practical benefits: fine-tuning a task-relevant pre-trained representation is fast, there is no need to store or revisit expert pre-training data. How is such a library of expert representations populated and maintained? Previous work (*e.g.*, Puigcerver et al. (2020); Achille et al. (2019); Deshpande et al. (2021)) has assumed a static collection of experts created by training on domain-specific datasets or domain-specific subsets of general dataset. Instead, we would like to automatically enrich the diversity of the expert library by accumulating knowledge from transferred downstream tasks, so that the overall system performance continually increases over time (life-long meta-learning).

One approach to growing the library would be to add fine-tuned downstream task-specific models back into the library as a candidate expert for transfer on future tasks. Such a naïve approach clearly does not scale and, at a minimum, requires developing techniques for selecting experts that is sub-linear in the size of the library (Puigcerver et al., 2020; Achille et al., 2019; Deshpande et al., 2021). More importantly, as our experiments show in Sec. 4, task-specific models fine-tuned on small amounts of data do not provide transferable representations. Task-specific models tend to overspecialize and degrade under-utilized features in their representations, and thus under-perform on new tasks compared to generic pre-trained models.

To address this, we introduce *representation consolidation* where the goal is to consolidate knowledge from multiple task-specific teacher models into a single expert student representation which transfers to downstream tasks *better* than any of the individual teacher representations. There are three factors that make the problem of representation consolidation unique relative to the existing literature: (1) we assume a representation is to be consolidated from a collection of *multiple* task-specific models, (2) we avoid the need to revisit task-specific training data of these models, and (3) we focus on improving downstream transfer instead of simply replicating performance on the upstream teacher's task(s). To the best of our knowledge, no prior work has analyzed the downstream transfer aspect of distilled domain-expert representations.

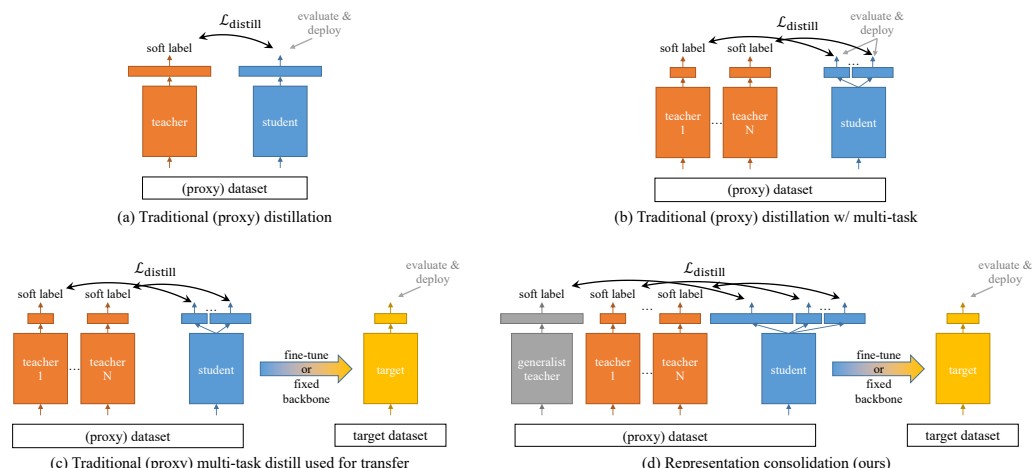

Figure 1: Knowledge distillation (a,b) seeks to copy the end-to-end functionality of one or more teachers but often fails to transfer (c). We propose a simple yet effective method to learn a consolidated representation from $N$ teachers that transfers well to downstream tasks (d). Given a large unlabeled proxy dataset, we train a single student model using multi-task distillation with a separate classifier head for each of the teacher tasks. To limit student forgetting and representation collapse, we always include an additional generalist teacher (ImageNet model). We show that the resulting consolidated representation transfers better to downstream tasks than any of the individual teachers (including the generalist).

To carry out representation consolidation, we utilize multi-teacher multi-task model distillation (see Fig. 1). We use a simple yet effective method to jointly distill one or several task-specific teacher with a generalist one. Each teacher may operate on a different set of classes, and a multi-head student is trained to emulate all teachers. Previous work on knowledge distillation has focused on the student model's performance on the teacher's task. Instead, we evaluate how well the student representation generalizes to *new* downstream tasks (whether related or unrelated to the teachers' tasks). In this new setting we demonstrate several surprising results:

- Task-specific model representations often transfer poorly even to a related downstream task.

- Consolidating a task-specific teacher with a generalist teacher (ImageNet) is sufficient to rescue the student. The resulting representation matches or improves the downstream performance over both task-specific and generalist representations, getting the best of both worlds even though no task-specific data is replayed.

- Consolidating multiple related task-specific teacher models can yield a better student representation that exceeds the performance of any one teacher on downstream tasks.

- Consolidation performs similarly to a multi-task joint training oracle, therefore gaining almost all of the transfer benefit from the teacher's dataset without needing to replay it.

| Symbol | Definition |
|---|---|
| $N$ | The number of tasks / task-specific teachers |
| $\mathcal{D}_t^i$ | The $i$-th task-specific teacher's dataset |
| $\phi_t^i$ | The $i$-th task-specific teacher's backbone ($i=0$: the generalist's backbone) |
| $h_t^i$ | The $i$-th task-specific teacher's head (classifier layer) ($i=0$: the generalist's head) |
| $\mathcal{D}_{\text{proxy}}$ | The large unlabeled proxy dataset used for distillation/consolidation |
| $\phi_s$ | The distilled/consolidated student's backbone |
| $h_s^i$ | The distilled/consolidated student's $i$-th head (classifier) |

| Symbol | Definition |
|---|---|
| $\mathcal{D}_d^j$ | The $j$-th **d**ownstream task's dataset |
| $\phi_d^j$ | The consolidated student backbone fine-tuned on dataset $\mathcal{D}_d^j$ |
| $h_d^j$ | The downstream model's head (classifier layer or linear SVM) for dataset $\mathcal{D}_d^j$ |
| $\lambda_i$ | Loss weight for the $i$-th teacher |
| t-split | First 50% random split of classes of a dataset, used as $\mathcal{D}_t^i$ |
| d-split | Second 50% random split of classes of a dataset, used as $\mathcal{D}_d^j$ |

Table 1: Glossary of symbols.

## 2 REPRESENTATION CONSOLIDATION

**Problem statement.** We start with a collection of one or more task-specific image classification models $\{\mathcal{M}_t^i\}_{i=1}^N$ as **t**eachers, trained on corresponding datasets $\{\mathcal{D}_t^i\}$ belonging to some domain (satellite images, images of flowers, *etc.*). We assume models consist of a feature extractor or backbone $\phi_t^i(\cdot)$, composed with a classifier head $h_t^i(\cdot)$ so that $\mathcal{M}_t^i = h_t^i(\phi_t^i(\cdot))$. We consolidate the knowledge of these task-specific teachers into a single **s**tudent representation $\phi_s(\cdot)$ using a proxy dataset $\mathcal{D}_{\text{proxy}}$ (*e.g.*, ImageNet), and then evaluate $\phi_s(\cdot)$ by training a SVM on top of it (or fine-tuning it) on a given **d**ownstream $\mathcal{D}_d^j$ chosen from some set $\{\mathcal{D}_d^j\}$. Our goal is that the resulting downstream model $h_d^j(\phi_d^j(\cdot))$ achieves good performance, where $\phi_d^j$ denotes the student $\phi_s$ as-is or potentially after tuning on $\mathcal{D}_d^j$. Fig. 1 highlights how this differs from standard distillation in which the student model $h_s(\phi_s(\cdot))$ is simply evaluated on the same task its teachers were once trained to perform.

**Forgetting and representation collapse during distillation.** Though our problem statement is novel, can existing approaches solve our problem statement? We observe (Sec. 4) that neither the task-specific teachers nor students distilled with standard or state-of-the-art methods provide sufficiently transferable representations. They under-perform general pre-training representations (*e.g.* with ImageNet) when evaluated on downstream tasks from the teachers' domain, and drastically under-perform on tasks outside the teachers domain. Surprisingly, this holds true *even though we use ImageNet-pretrained weights to initialize the teacher and student networks*. Thus, we argue that training task-specific teachers or distilling from them suffers from catastrophic forgetting of general knowledge that is crucial for transfer learning.

Intuitively, transfer performance depends on how distinguishable different classes are represented in the penultimate layer feature space. In standard distillation, the student only learns from task-specific teachers trained on smaller datasets and is only required to discriminate the classes in those datasets. This is well suited for the traditional distillation problem that evaluates the student on these same tasks. But for representation consolidation, distinguishing *unknown downstream* classes requires preserving general features that may not be relevant to the teachers' specific task. Our strategy is thus to ensure that the student maintains general features as it learns task-specific ones.

**Method formulation.** A key challenge to our approach is avoiding replay of the images used to train the task-specific teachers. To overcome this, we propose using a proxy dataset and a generalist model that transfers well (*e.g.* one pre-trained on ImageNet). This model is referred to as a generalist as it is not task specific and contains a wide variety of classes, and use it to help avoid forgetting. More specifically, we use multi-head multi-task distillation. We put $N + 1$ heads, $h_s^0, \ldots, h_s^N$, on top of the student backbone $\phi_s(\cdot)$. In addition to the task-specific teachers $h_t^i(\phi_t^i(\cdot))$, $i \in \{1, \ldots, N\}$ used in traditional distillation, we also include the generalist teacher trained on ImageNet (denoted as $h_t^0(\phi_t^0(\cdot))$). The rationale behind this construction is that we learn task knowledge from the task-specific model on the proxy dataset, but also force the student to retain knowledge from the generalist teacher to prevent over-specialization. We learn the student by optimizing the loss:

$$\mathcal{L} = \sum_{x \in \mathcal{D}_{\text{proxy}}} \left( \lambda_0 \underbrace{\mathcal{L}_{\text{distill}} \left( h_t^0(\phi_t^0(x)), \; h_s^0(\phi_s(x)) \right)}_{\text{generalist teacher term}} + \sum_{i=1}^N \lambda_i \underbrace{\mathcal{L}_{\text{distill}} \left( h_t^i(\phi_t^i(x)), \; h_s^i(\phi_s(x)) \right)}_{\text{traditional distillation}} \right) \quad (1)$$

with a distillation loss $\mathcal{L}_{\text{distill}}$. Our key difference from traditional distillation is the generalist teacher term. Setting $\lambda_0 = 0$ while keeping all other conditions (proxy dataset, network initialization, etc.) yields a standard multi-teacher distillation baseline. Note that this is also distinct from ensemble knowledge distillation, where one averages the homogeneous output of every model and distills the mean onto a single student head, since our teacher outputs are heterogeneous (different tasks).

Our method is agnostic to the underlying distillation method. We can use the standard knowledge distillation (KD) loss (Hinton et al., 2015) which is cross-entropy with temperature $T = 2$, *i.e.* $\mathcal{L}_{\text{distill}}(p_t, p_s) = -\sum_{c=1}^C p_t^{(c)} \log(p_s^{(c)})$ where $c$ indexes the $C$ classes, and $p_t = \text{softmax}\left(h_t^i(\phi_t^i(x))/T\right)$, $p_s = \text{softmax}\left(h_s^i(\phi_s(x))/T\right)$. Alternatively, we can use a state-of-the-art distillation method, *e.g.*, CRD (Tian et al., 2020). CRD uses the KD loss and adds a contrastive embedding loss between $h_{\text{CRD,t}}^i(\phi_t^i(x))$ and $h_{\text{CRD,s}}^i(\phi_s(x))$ with a pair of trainable linear embedding layers $h_{\text{CRD,t}}^i$ and $h_{\text{CRD,s}}^i$ that are used only in training and discarded afterwards. The teacher $\phi_t^i$'s

are trained on completely different tasks, so for CRD we use different embedding layers for each teacher-student-head pair.

We initialize the student backbone $\phi_s$ and its 0-th head $h_s^0$ using the generalist model's weights $\phi_t^0, h_t^0$, whereas other heads are randomly initialized. Since it is important to maintain pre-trained model's representational power, we simply set the loss weights $\lambda_0 = 1$ and $\lambda_i = \frac{1}{N}$ for $1 \leq i \leq N$. This forces the learned representation $\phi_s$ to balance between learning general and task-specific features, which benefits future downstream transfer. After training the student we evaluate the resulting representation $\phi_s(\cdot)$ on multiple downstream tasks $\{\mathcal{D}_d^i\}$. For each task $j$ we can either fine-tune the whole model $h_d^j(\phi_d^j(\cdot))$ or keep the student representation fixed and only learn the classifier head $h_d^j(\phi_s(\cdot))$ which is often referred to as a *linear probe*.

## 3 EXPERIMENTAL SETUP

**Datasets and downstream tasks.** We utilize datasets from a variety of domains to generate teachers and downstream tasks: Cars196 (Krause et al., 2013), Resisc45 (Cheng et al., 2017) (remote sensing images), iFood (Kaur et al., 2019) and Food101 (Bossard et al., 2014), iFashion (Guo et al., 2019), DTD (Cimpoi et al., 2014) (describable textures), iNaturalist (Horn et al., 2018) (species classification, 2019 challenge version), CUB Birds (Wah et al., 2011), Flowers (Nilsback & Zisserman, 2008), Caltech256 (Griffin et al., 2007), and Aircrafts (Maji et al., 2013). Among these, iFood and Food101 are the same domain, and Birds and Flowers are subdomains of iNaturalist. We checked for near-duplicates between these datasets using perceptual hash (Buchner, 2021), and found negligible duplication: 1 out of 134k iNaturalist (50% classes we used) is a duplicate of CUB, and 8 out of 130k/100k images are duplicates between iFood and Food101.

To evaluate if a consolidated student representation has learned features relevant to a specific domain, we require downstream tasks that are related (in the same domain) to that of each task-specific teacher. Except for Food101 and iFashion, we split each dataset at random into two disjoint sets which each contain only 50% of the classes. We take the first half of the dataset, named "t-split", and use as $\mathcal{D}_t^i$, to train a task-specific teacher. We use the second half of each dataset, named "d-split", as one of the downstream tasks $\mathcal{D}_d^j$. We primarily evaluate on few-shot downstream transfer, where we randomly sample 5 training images from each available class in $\mathcal{D}_d^j$, but always use the entire test set for evaluation. We use all samples for those classes with $< 5$ images. For iFashion, which is a multi-task multi-label dataset, we use all images but with a subset of labels (*e.g.*, those related to clothing category) as $\mathcal{D}_t^i$ for teacher training. We then use all images with a disjoint set of labels (*e.g.*, those related to sleeve style) as a downstream task $\mathcal{D}_d^j$ for evaluating transfer. For iFashion's few-shot scenario, we randomly subsample 1000 images for downstream training, and evaluate on all test data. We do not train any teacher on Food101 so the complete set of classes are used as a downstream task.

During distillation and consolidation, we use ImageNet (Russakovsky et al., 2015) (ILSVRC12) as the proxy dataset $\mathcal{D}_{\text{proxy}}$ for most experiments. However, we also show results with Places365-standard (Zhou et al., 2018) or iNaturalist (Horn et al., 2018) as the proxy.

**Compared methods and Criteria.** As a reminder, our goal is not to duplicate upstream teacher predictions, but rather to improve transfer performance of distilled representations. Therefore, we evaluate each student's $\phi_s$ using its performance when transferred to various downstream datasets $\{\mathcal{D}_d^j\}$ and compare multiple methods for initializing the downstream representation $\phi_d^j$:

(1) ImageNet-pretrained $\phi_t^0$, a strong baseline for transfer learning. (2) ImageNet-pretrained $\phi_t^0$ fine-tuned on the soft-labels produced by the ImageNet-pretrained model with batchnorms in test mode. This baseline aims to isolate the effect of soft-labels / self-distillation on model performance. (3) The task-specific teacher $\phi_t^1$ (or one of the teachers when $N > 1$) without further distillation. (4) Distillation with $N$ teachers using either standard knowledge distillation (KD) or contrastive representation distillation (CRD). This distills only task-specific teacher(s) on $\mathcal{D}_{\text{proxy}}$, without the ImageNet teacher $h_t^0(\phi_t^0(\cdot))$. (5) Our consolidated representation $\phi_s$ which includes $h_t^0(\phi_t^0(\cdot))$ as a teacher, using either KD or CRD as the underlying distillation loss. (6) A multi-task learning oracle (MTL) that jointly trains on ImageNet and teacher data $\mathcal{D}_t$ rather than on unlabeled proxy data.

| Dataset | iFood (t-split) | iFood (d-split) 5-shot | Food101 5-shot | Resisc45 (t-split) 5-shot |
|---|---|---|---|---|
| evaluated head | $h_t^1$ or $h_s^1$ | SVM $h_d$ | SVM $h_d$ | SVM $h_d$ |
| ImageNet pre-train | – | 28.9 | 37.0 | 70.7 |
| Teacher (t-split) | 74.4 | 34.8 | 42.8 | 53.3 |
| KD (trad. distill) | 72.8 | 35.3 | 44.0 | 53.6 |
| Ours + KD | 68.6 | 38.8 | 47.2 | 69.5 |
| MTL (oracle) | 71.6 | 38.9 | 46.7 | 69.4 |

(a) iFood as task-specific teacher dataset

| Dataset | Resisc45 (t-split) | Resisc45 (d-split) 5-shot | iFood (d-split) 5-shot | Food101 5-shot |
|---|---|---|---|---|
| evaluated head | $h_t^1$ or $h_s^1$ | SVM $h_d$ | SVM $h_d$ | SVM $h_d$ |
| ImageNet pre-train | – | 70.7 | 28.9 | 37.0 |
| Teacher (t-split) | 98.2 | 67.9 | 14.8 | 17.9 |
| KD (trad. distill) | 97.9 | 61.6 | 9.9 | 12.1 |
| Ours + KD | 97.0 | 72.6 | 28.8 | 36.4 |
| MTL (oracle) | 96.9 | 73.1 | 29.1 | 36.2 |

(b) Resisc45 as task-specific teacher dataset

Table 2: **Exp. 0:** Accuracies of baselines and representation consolidation, as judged with traditional criteria (first column, original network head's old task performance) and transfer learning criteria (last 3 columns, performance of linear SVM head trained on related (bold) & unrelated downstream tasks). Baselines work well for the original task, but underperform in transfer learning. Ours matches or outperforms the best of all baselines in all transfer scenarios and matches the oracle.

For comparison fairness, (2-6) are all ImageNet pre-trained (*i.e.* initialized with (1)) before further training. Both (4) and (5) are using the same proxy dataset (*i.e.* ImageNet for most experiments).

We use the transfer accuracy on downstream tasks $\mathcal{D}_d^j$ to measure each representation's power. We primarily use the few-shot linear probe (train a 5-shot linear SVM as $h_d^j$ over fixed $\phi$) on $\mathcal{D}_d^j$'s training set (single training run for full dataset, few-shot performance averaged over 50 random subsampling trials). We also verify our results hold when fine-tuning the student representation $h_d^j(\phi_d(\cdot))$ on few-shot or full dataset splits.

**Implementation details.** We will release our code to reproduce this paper upon acceptance. For experimental fairness, we use the same network (ResNet50 (He et al., 2016)) / hyperparameters / ImageNet initialization for the baselines and ours. For more details, please see appendix A.

## 4 RESULTS

**Exp. 0: Motivational analysis – traditional distill *vs.* representation consolidation** To highlight the difference between representation consolidation and traditional distillation, we use either iFood or Resisc45 (t-split) as $\mathcal{D}_t^1$ to train teachers, and run distillation/consolidation with $N = 1$. We test on iFood (d-split), Food101 (full), and Resisc45 (d-split) as downstream tasks (t-split and d-split are disjoint 50% classes of each dataset; see Section 3)

On downstream tasks $\mathcal{D}_d^j$, we follow representation learning's evaluation protocol (few-shot linear probe): train linear SVM $h_d$ on $\phi(x)$, $x \in \mathcal{D}_d^j$, evaluate on $\mathcal{D}_d^j$'s test set. We also evaluate each network following the traditional distillation evaluation protocol, *i.e.* directly evaluate $h_t^1(\phi_t^1(\cdot))$ or $h_s^1(\phi_s(\cdot))$ on the upstream task $\mathcal{D}_t^1$'s test set. The results are shown in Table 2.

If we only focus on the upstream task $\mathcal{D}_t^1$, then traditional (proxy) distillation almost matches the teacher's performance, and representation consolidation performs worse than both. However, if we instead focus on the downstream transfer performance on $\mathcal{D}_d^j$, we see the opposite trend. In Table 2a with iFood representations, for the teacher-related downstream tasks Food101 and iFood d-split (the disjoint classes split from the same dataset as $\mathcal{D}_t^1$), we see a benefit of consolidation over both task-specific teacher and generic pre-trained features. On the unrelated downstream task (Resisc45), where the generalist teacher excels, we notice that our method almost matches the generalist performance even though we improve over it for the food tasks. We see a similar trend in Table 2b: the consolidated student outperforms the teacher and the generalist on the teacher-related task (Resisc45 d-split) but still retains the generalist teacher's performance on unrelated food tasks. This clearly demonstrates the significant difference between upstream and downstream transfer.

We perform nearly the same as the MTL oracle which trains also on the teacher datasets $\mathcal{D}_t$, indicating that while we avoid replaying $\mathcal{D}_t$, our simple consolidation method with $\mathcal{D}_{proxy}$ can extract nearly all of its benefit. Note that although we have carefully optimized the MTL oracle for downstream performance, adding state-of-the-art MTL techniques could outperform our MTL implementation.

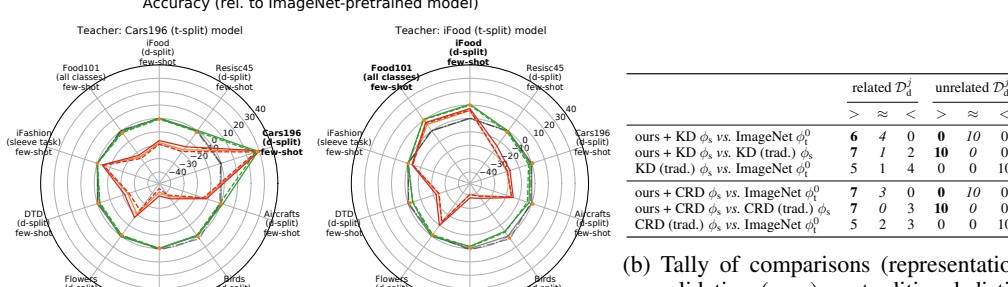

| | | related $\mathcal{D}_d^j$ | | | unrelated $\mathcal{D}_d^j$ | | |
|---|---|---|---|---|---|---|---|
| | | > | ≈ | < | > | ≈ | < |
| ours + KD $\phi_s$ vs. ImageNet $\phi_t^0$ | | **6** | *4* | 0 | **0** | *10* | 0 |
| ours + KD $\phi_s$ vs. KD (trad.) $\phi_s$ | | **7** | *1* | 2 | **10** | *0* | 0 |
| KD (trad.) $\phi_s$ vs. ImageNet $\phi_t^0$ | | 5 | 1 | 4 | 0 | 0 | 10 |
| ours + CRD $\phi_s$ vs. ImageNet $\phi_t^0$ | | **7** | *3* | 0 | **0** | *10* | 0 |
| ours + CRD $\phi_s$ vs. CRD (trad.) $\phi_s$ | | **7** | *0* | 3 | **10** | *0* | 0 |
| CRD (trad.) $\phi_s$ vs. ImageNet $\phi_t^0$ | | 5 | 2 | 3 | 0 | 0 | 10 |

(b) Tally of comparisons (representation consolidation (ours) *vs.* traditional distill *vs.* ImageNet pre-trained model) among all 10 teacher dataset $\mathcal{D}_t^1$ scenarios.

(a) Excerpt (2 of 10 $\mathcal{D}_t^1$ scenarios). Full figure in appendix Fig. 6.

Figure 2: **Exp. 1** with $N = 1$, 5-shot linear SVM downstream transfer. **Left:** Comparing different representations on ten downstream tasks. Teacher-related downstream tasks in **bold**. Performance relative to ImageNet representation baseline. Excerpt (2 of 10 teacher domain scenarios). **Right:** Tally of comparisons among all ten $\mathcal{D}_t^1$ domains. On teacher-related downstream tasks, we outperform or match ImageNet-pretrained, and on other tasks we match ImageNet. Traditional distill often underperforms ours (7/10 related, 10/10 unrelated) and ImageNet (3-4/10 related, 10/10 unrelated). See appendix for full results.

We only demonstrate that it is not trivial to outperform our method using original teacher training data.

**Exp. 1: Improving student representation when $N = 1$.** We show that this advantage of consolidation over baselines holds for a wide range of upstream and downstream datasets. We also compare to CRD to show state-of-the-art distillation often underperforms even when specifically targeting representation learning. Fig. 2 summarizes few-shot SVM accuracy using different representations relative to ImageNet-pretrained (See appendix for the full Fig. 6 and raw numbers).

The conclusions are similar regardless of using KD or CRD – consolidation outperforms (Cars196, Resisc45, iFood, CUB, Aircrafts, iNaturalist teachers) or matches (iFashion, DTD, Flowers, Caltech256 teachers) ImageNet pre-trained model performance on related downstream tasks, and matches its performance on unrelated ones. In contrast, both traditional distillation methods (1) underperform consolidation on related downstream tasks for all teachers except Cars196, iFashion, and Aircraft, and (2) drastically underperform both consolidated and ImageNet features on unrelated downstream tasks – although CRD is better than KD, it still degrades performance. Notably, traditional distillation methods underperform ImageNet even on related downstream tasks for Resisc45, DTD, Caltech256, and (for KD only) Flowers teachers. We match the MTL oracle performance across the board, reaping the teacher dataset's benefit without replaying it.

**Exp. 2: Consolidating representations with $N > 1$.** We can merge multiple task-specific teacher representations that are related to get a better one. In addition, our method is not constrained to merging models with the same architecture or $\phi_t^i$'s feature space dimensions like Geyer et al. (2019). To illustrate this, we split the "t-split" of Cars196 and iFood into five random splits containing 10% of the original classes. We train a ResNet18 teacher on each of the five splits. Then, we use either traditional distillation or representation consolidation to merge these models with a ResNet50 generalist teacher. We compare the resulting representations with the teacher model's and ImageNet. Fig. 3a shows this result. For downstream tasks, we obtain better performance than using only one of the five teachers on both related and unrelated tasks, especially for iFood and Food101 where the teachers themselves underperform on related downstream task. This shows that our method can benefit from even teachers whose representation is weaker, as long as they have domain knowledge.

Full comparison with traditional distillation and representation consolidation from only one of the five teachers are in the appendix – we gain performance on similar $\mathcal{D}_d^j$ by using five task-specific teachers instead of one, and we outperform both traditional distill methods on related downstream

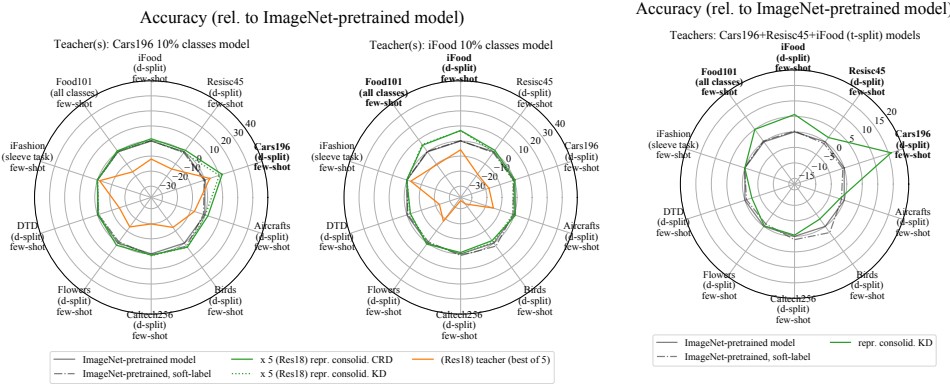

(a) Merging same-domain ResNet18 teachers

(b) Merge multi-domain Res50 teachers

Figure 3: **Exp. 2** with $N > 1$ multi-model merging, 5-shot linear SVM downstream transfer. **Left:** In the same domain, we are able to consolidate from models with different architectures and improve transfer performance over every single teacher. **Right:** We can consolidate different domain models and improve over the ImageNet representation. See appendix for full comparisons.

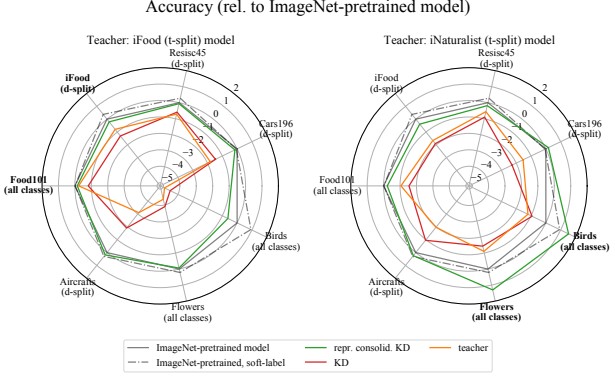

Figure 4: **Exp. 3** Excerpt of $N = 1$, fine-tuning results. Showing full-shot (transfer to all classes for Flowers and Birds). Our conclusions are the same between fine-tuning and fixed representation (Fig. 2). See appendix for the full results (few-shot, full-shot, more datasets).

tasks for iFood/Food101. We also show results when using all ResNet50 teachers for the five splits in the appendix. The gaps are closer but the results are similar and the conclusions are identical.

Finally, we explore merging models from different domains to form a multi-domain consolidated student. See Fig. 3b. We observe that we can outperform ImageNet on all related downstream tasks, but the performance gain is smaller than representation consolidation in just one domain. We show in the appendix that ours+KD outperforms KD on most downstream tasks except Cars196.

**Exp. 3: Fine-tuning downstream.** We also verify that our conclusions generalize to fine-tuning as well, especially without few-shot sampling. Fig. 4 shows an excerpt of our results that include full dataset transfer. See appendix for fine-tuning using few-shot and full d-splits, whose conclusions are the same as this section's. Fine-tuning allows the representation to change into one that better suits the downstream task, so the benefit of teachers' domain knowledge shrinks compared to fixed $\phi_s$ with linear SVM $h_d^j$. Despite this, the conclusions are the same as the fixed $\phi_s$ scenario.

**Exp. 4: Influence of the Loss weights.** Fig. 5a shows the effect of the choice of loss weight on our method – when we use $\lambda_0 = \frac{1}{2}, \lambda_i = \frac{3}{2N}$ $(i > 0)$, we gain performance on the related downstream task but lose performance on unrelated ones, whereas using $\lambda_0 = \frac{3}{2}, \lambda_i = \frac{1}{2N}$ $(i > 0)$ gives us the opposite. We can potentially use this trade-off to suppress unreliable teachers' influence.

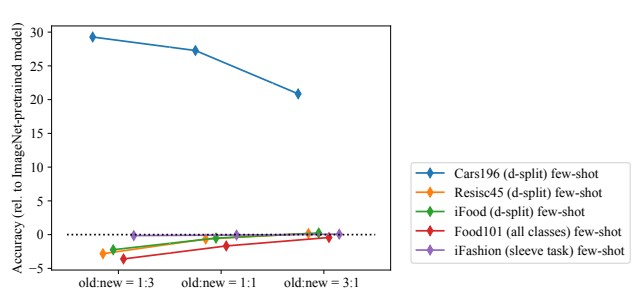

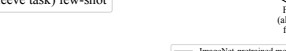

(a) **Exp. 4** Tune loss weights $\lambda_0$ (old) and $\lambda_1$ (new). $\mathcal{D}_t^1 = $ Cars196.

(b) **Exp. 5** $\mathcal{D}_{\text{proxy}}$. More in appendix.

Figure 5: Additional studies. 5-shot linear SVM downstream transfer. **Left:** Tuning loss weights result in different balance between related and unrelated downstream task performance. **Right:** Places365 as proxy has similar results as ImageNet proxy. However, training on Places365's supervised labels or using a less general proxy (iNaturalist) hinders performance.

**Exp. 5: Proxy data choice.** Fig. 5b shows results replacing ImageNet with Places365 or iNaturalist as the $\mathcal{D}_{\text{proxy}}$ dataset. Places365 yields similar performances as using ImageNet, while a narrower-scope iNaturalist is weaker on unrelated downstream tasks. Jointly training on Places365 labels is not helpful, meaning that the improvement comes from the generalist teacher, not the data.

## 5 RELATED WORK

Our goal is to maximize the *transferability* of a representation consolidated from multiple teachers. This downstream transfer aspect has received little attention in the literature (Liu et al., 2019; Geyer et al., 2019; Tian et al., 2020), but this problem formulation is closely related to prior work on multi-model merging and distillation with proxy data.

**Multi-model merging.** It is useful to combine separate models that perform different tasks into a single model for efficiency and performance benefits. Knowledge Concentration (Gao et al., 2017) combine teachers trained on subsets of 100k classes in the EFT dataset (Gao et al., 2017) into one improved single model, using handcrafted sparse connections for the final student layers. Chou et al. (2018) merge CNNs by combining the kernel weights of different models by clustering and lookup, followed by fine-tuning. Zhao et al. (2020) merge object detection models when some classes may be the background to other models. Vongkulbhisal et al. (2019) combine models with overlapping classes using the combined train set, by deriving soft labels from intra-model class correspondence. Ye et al. (2020) progressively train a GAN to regenerate proxy data for all teachers, and combine teachers layer by layer. Chakraborty et al. (2018) and Park & Kwak (2020) aggregate ensembles of members trained on the same task. In one-shot federated learning (Guha et al., 2019), multiple clients with their own private data train a model each, and finally merge the models using an ensemble or distillation in a way that protects their data privacy.

Unlike our approach, these methods merge models in order to perform exactly the *same task* as the teacher models, are not concerned with the performance of the student when transferred to a downstream task, and often (Gao et al., 2017; Chou et al., 2018; Zhao et al., 2020; Vongkulbhisal et al., 2019; Chakraborty et al., 2018; Park & Kwak, 2020) require revisiting the original training images. We have shown that these teachers themselves have poor transferability compared to a simple pre-trained baseline, and learning only from them (even using state-of-the-art CRD (Tian et al., 2020)) yields suboptimal transfer learning performance.

**Multi-model merging for transfer.** Knowledge Flow (Liu et al., 2019) connects the student to multiple teachers' intermediate layers to kick-start target task training, and gradually penalizes its reliance on teacher models over time. Geyer et al. (2019) uses IMM (Lee et al., 2017) to merge multiple models using their diagonal approximated Fisher information matrix (FIM) to balance different

teacher's weight importance, merge, and fine-tune. Computing the FIM requires reprocessing the original teacher training data (unlike our approach that only needs generic proxy data). Furthermore, this method requires homogeneous architectures for students and teachers, while ours works on any architecture combinations. These methods directly optimize performance on the downstream task and thus require re-running for each different target dataset. Our approach is more efficient, as it only requires consolidating teachers once to improve the pre-trained representation independently of the downstream task. Finally, we note that the representations learned were not compared to a strong baseline (i.e., pre-training on ImageNet), which we argue is a prerequisite for being useful in real-world applications. One of our main contributions is observing the need for including a generalist teacher, an insight which is orthogonal to, and could be combined with these previous approaches.

**Distillation with representation losses ("representation distillation")** tries to capture additional structure of feature representations by aligning student and teacher feature activations during distillation. Koratana et al. (2019) compress models by adding $L_2$ losses between intermediate representations of the teacher and students. Aguilar et al. (2020) use KL divergence and cosine similarity to make the attention and representation of intermediate features of the student and teacher similar. Tian et al. (2020) adds contrastive representation learning loss to the penultimate layer to preserve feature structures, by maximizing each image's student and teacher features' mutual information. Despite their focus on yielding better representations, these methods fully rely on the teacher's transferability. We show that this yields downstream performance worse than generalist baselines.

**Proxy data ("data-free") distillation** transfers the input-output function of a teacher network to a student network without using the teacher's training data. (Yalniz et al., 2019; Orekondy et al., 2019) opt to use a large general proxy dataset to query the teacher, and their teacher outputs to on this data to train the student. Other methods (Nayak et al., 2019; Chen et al., 2019; Haroush et al., 2020; Chawla et al., 2021) generate proxy data directly from the trained models, and use this data to train the students. Further, Micaelli & Storkey (2019); Yin et al. (2020) also encourage generating samples the student and teacher disagree on. Lastly, other methods require the original dataset to compute meta-data information such as feature cluster mean. (Lopes et al., 2017; Bhardwaj et al., 2019) Some train a GAN from the teachers to maximize chosen class predictions (Fang et al., 2019; Yoo et al., 2019; Ye et al., 2020), sometimes also batchnorm statistics (Luo et al., 2020; Xu et al., 2020b), and sometimes on proxy data instead. (Addepalli et al., 2020; Besnier et al., 2020) Some work combines self-supervised learning with distillation (Tian et al., 2020; Xu et al., 2020a; Fang et al., 2021) to improve the representation performance on the original task.

These works try to emulate the original data for standard distillation, and do not concern the student's transferability or merging multiple teachers. Our simple method already performs comparably to a multi-task oracle that uses the original teacher dataset, indicating that these more complex methods that improve upstream task distillation may be unnecessary when the goal is downstream transfer.

**Incremental learning** is also related to our overall goal of growing a library of expert representations. These methods continually learn tasks or classes but often with limited access to past training data (Li & Hoiem, 2018; Rebuffi et al., 2017; Kirkpatrick et al., 2017; Zenke et al., 2017; Hu et al., 2019; Yin et al., 2020; Prabhu et al., 2020). Our approach addresses many of the same challenges by consolidating knowledge in the form of feature representations without revisiting old data used to train teachers, but our "increments" are whole tasks given in the form of trained models instead of data, whose labels may overlap. **Multi-task learning** (Caruana, 1998) assumes all datasets are available and trains jointly on them whereas we assume task-specific datasets absent. We perform comparably to this oracle.

## 6 SUMMARY

In this paper, we show that traditional distillation can result in a representation suboptimal for downstream task transfer learning, because it only focuses on preserving the end-to-end input-output mapping of the old task. We propose representation consolidation with the generalist model as an additional teacher. Our method preserves the wide-range transferability of the strong ImageNet baseline and improve the performance for both related and unrelated downstream tasks over traditionally distilled networks. We show that we can merge multiple models in the same domain to get a better representation than any single model.

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

## A    IMPLEMENTATION DETAILS

We will release our code to reproduce this paper upon acceptance. We use PyTorch. (Paszke et al., 2019) For all network training, we use SGD with momentum of 0.9, weight decay of $10^{-4}$, a batch size of 32. Unless noted, we use a learning rate decay of $0.1$ at 50% and 80% of total training. We initialize task-specific teachers' training and all students' distillation with a ResNet50 (He et al., 2016) pre-trained on ImageNet, and we freeze batchnorm statistics during distill/consolidation/MTL. We learn each $\phi_t^i$ on the task-specific $\mathcal{D}_t^i$ by fine-tuning 120 epochs (learning rate decay at 70th and 100th epoch) while doing a log-scale grid search on the learning rate. For distilling $\phi_s$, our method is less sensitive to the choice of learning rate. We use a fixed learning rate of 0.01 for each $h_s^i$ and 0.001 for $\phi_s$, and a schedule of 40 epochs. Note that ImageNet pre-training in PyTorch uses 0.001 as the final epoch learning rate. This takes us roughly 4 days on an AWS instance with one NVIDIA V100 for each distilled/consolidated representations. When the downstream transfer uses a fixed $\phi_d^j$, we extract $\phi_d(x)$ on the center image crop, and search $h_d^j$'s SVM hyperparameters using a 5-fold cross-validation in scikit-learn (Pedregosa et al., 2011). When the downstream transfer uses fine-tuning, we run a log-scale grid search of learning rate with a 50 epoch schedule.

## B    COMPLETE FIGURES FOR SECTION 4 EXPERIMENTS

We now provide the full graphs and table for our experiments. Note that we have summarized these results and all conclusions in the main paper.

**Exp. 1: Improving student representation when $N = 1$.** Fig. 6 and Table 3 show full results for Fig. 2. Please see the main paper for analysis and conclusions.

**Exp. 2: Consolidating representations with $N > 1$.** Figs. 7, 8, 9 shows full results for Fig. 3.

For Figs. 7 and 8 same-domain model merging, in addition to the main paper results merging ResNet18 $\phi_t^i$ and ResNet50 $\phi_t^0$, we show similar results for all teachers being ResNet50 in Fig. 8. We also show comparison to traditional distill with 5 teachers, representation consolidation with only 1 teacher, and the teacher itself (best out of 5 according to related $\mathcal{D}_d^j$ performance). The conclusions are the same, and merging five teachers using representation consolidation outperforms all baselines on both related and unrelated downstream tasks (except Cars196 with related $\mathcal{D}_d^j$ against traditional distill).

For multi-domain model merging in Fig. 9, we compare to traditional distill with multi-task learning. We outperform or match it on downstream $\mathcal{D}_d^j$ except for Cars196. We also explore using a concatenation of multiple large unlabeled datasets as $\mathcal{D}_{\text{proxy}}$. With a more diverse proxy, the performance

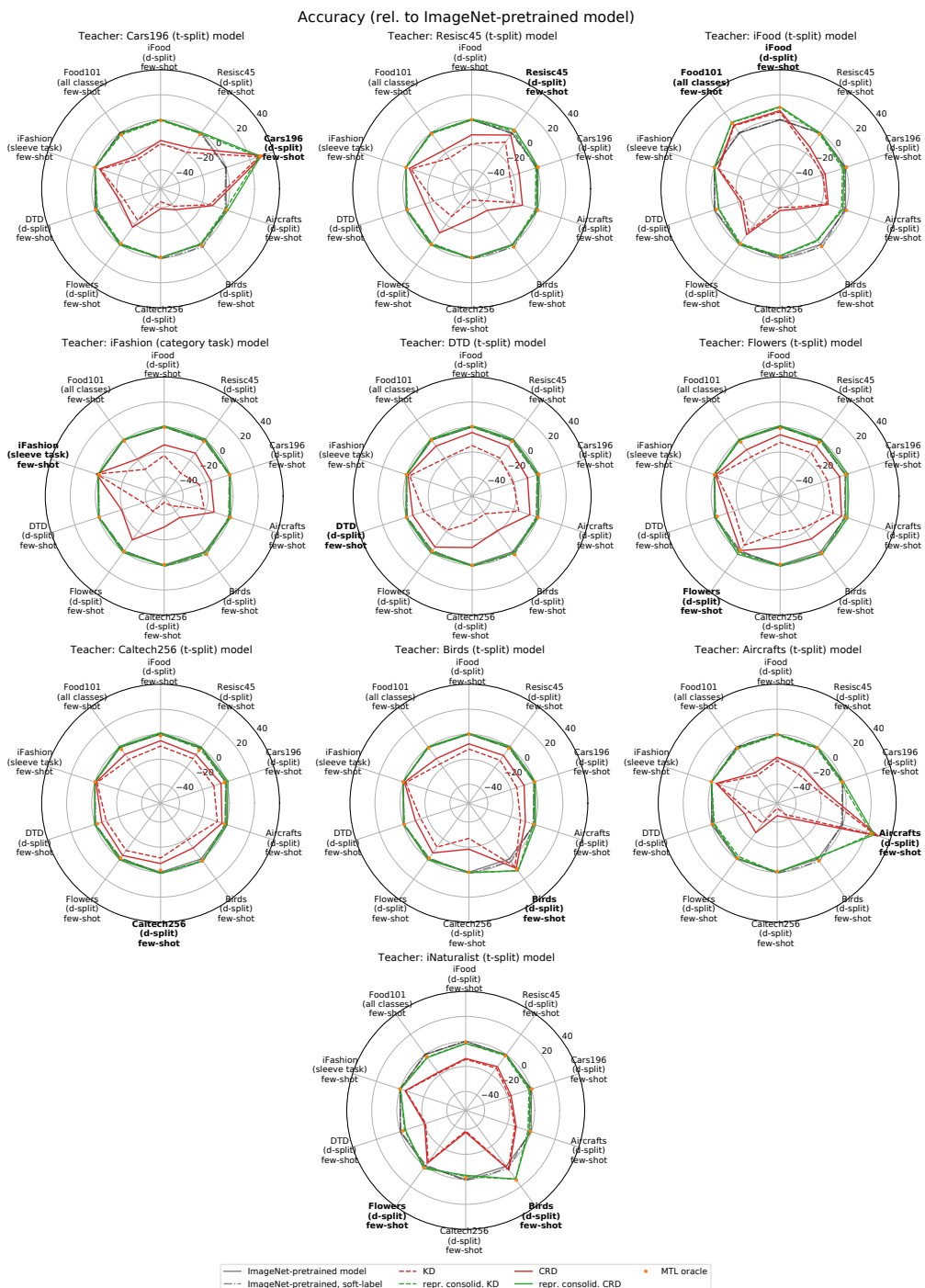

Figure 6: **Exp. 1** Full results for Fig. 2a ($N = 1$ single task-specific teacher, 5-shot linear SVM downstream transfer). All 10 $\mathcal{D}_t^i$ cases. This extensive set of experiments have the same conclusions as the main paper: we match or outperform ImageNet representation while traditional distill often underperforms. See also Table 3 for a comparison tally.

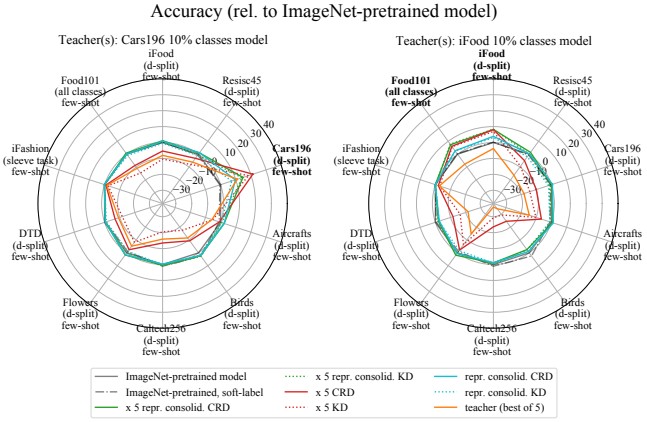

Figure 7: **Exp. 2** Merging $N > 1$ same-domain ResNet18 teachers, 5-shot linear SVM downstream transfer. Part 1/2 of full results for Fig. 3a. We are able to consolidate from models with different architectures (ResNet50 $\phi_t^0$ and ResNet18 $\phi_t^i$) and improve transfer performance over every single teacher.

Figure 8: **Exp. 2** Merging $N > 1$ same-domain ResNet50 teachers, 5-shot linear SVM downstream transfer. Part 2/2 of full results for Fig. 3a. Our conclusions generalize to using all ResNet50 teachers.

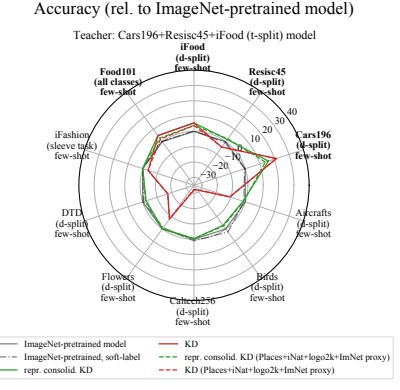

Figure 9: **Exp. 2** Merging $N > 1$ different domain ResNet50 teachers, 5-shot linear SVM downstream transfer. Full results for Fig. 3b ($N > 1$ multiple model merging, multiple domains). We can consolidate different domain models and improve over the ImageNet representation and (for most related/unrelated downstream datasets) multi-task traditional distillation.

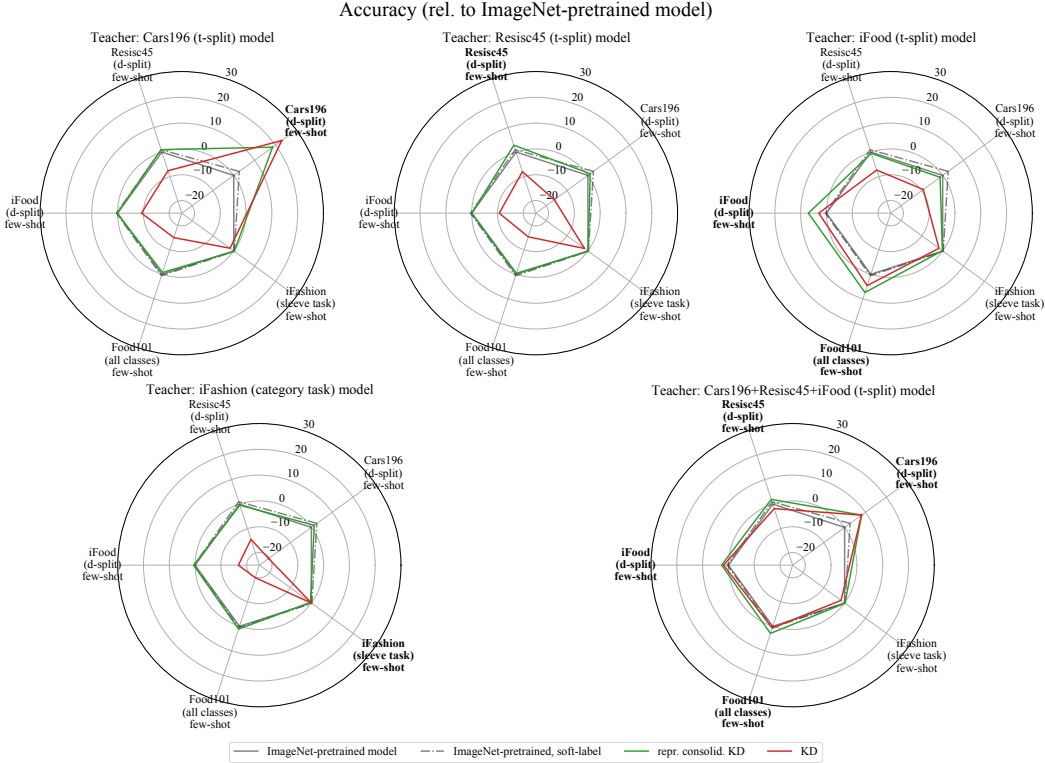

Figure 10: **Exp. 3** $N = 1$ (first 4 graphs) and $N > 1$ (last graph) teacher(s), 5-shot **fine-tuning** downstream transfer. Part 1/2 of full results for Fig. 4. The same conclusions as the fixed representation scenario in Figs. 2, 6 hold.

drops a little for related $\mathcal{D}_d^j$ and stays similar for unrelated $\mathcal{D}_d^j$, suggesting we are somewhat insensitive to choice of datasets, but a more diverse proxy data may not provide better model merging performance.

| | related $\mathcal{D}_d^j$ | | | unrelated $\mathcal{D}_d^j$ | | |
|---|---|---|---|---|---|---|
| | > | ≈ | < | > | ≈ | < |
| ours + KD $\phi_s$ vs. ImageNet $\phi_t^0$ | (6 others) | iFashion, DTD, Flowers, Caltech256. | | | (all 10) | |
| ours + KD $\phi_s$ vs. KD (trad. distill) $\phi_s$ | (7 others) | Cars196. | iFashion, Aircraft. | (all 10) | | |
| KD (trad. distill) $\phi_s$ vs. ImageNet $\phi_t^0$ | (5 others) | iNaturalist. | Resisc45, DTD, Flowers, Caltech256. | | | (all 10) |
| ours + CRD $\phi_s$ vs. ImageNet $\phi_t^0$ | (7 others) | iFashion, DTD, Caltech256. | | | (all 10) | |
| ours + CRD $\phi_s$ vs. CRD (trad. distill) $\phi_s$ | (7 others) | | Cars196, iFashion, Aircraft. | (all 10) | | |
| CRD (trad. distill) $\phi_s$ vs. ImageNet $\phi_t^0$ | (5 others) | Flowers, iNaturalist. | Resisc45, DTD, Caltech256. | | | (all 10) |

Table 3: **Exp. 1** Detailed tally of Fig. 6 ($N = 1$ single task-specific teacher, 5-shot linear SVM downstream transfer) explaining Fig. 2b. On teacher-related downstream tasks, we outperform or match ImageNet, and on other tasks we match ImageNet performance. Traditional distill often underperforms ours (7/10 related, 10/10 unrelated) and ImageNet (3-4/10 related, 10/10 unrelated).

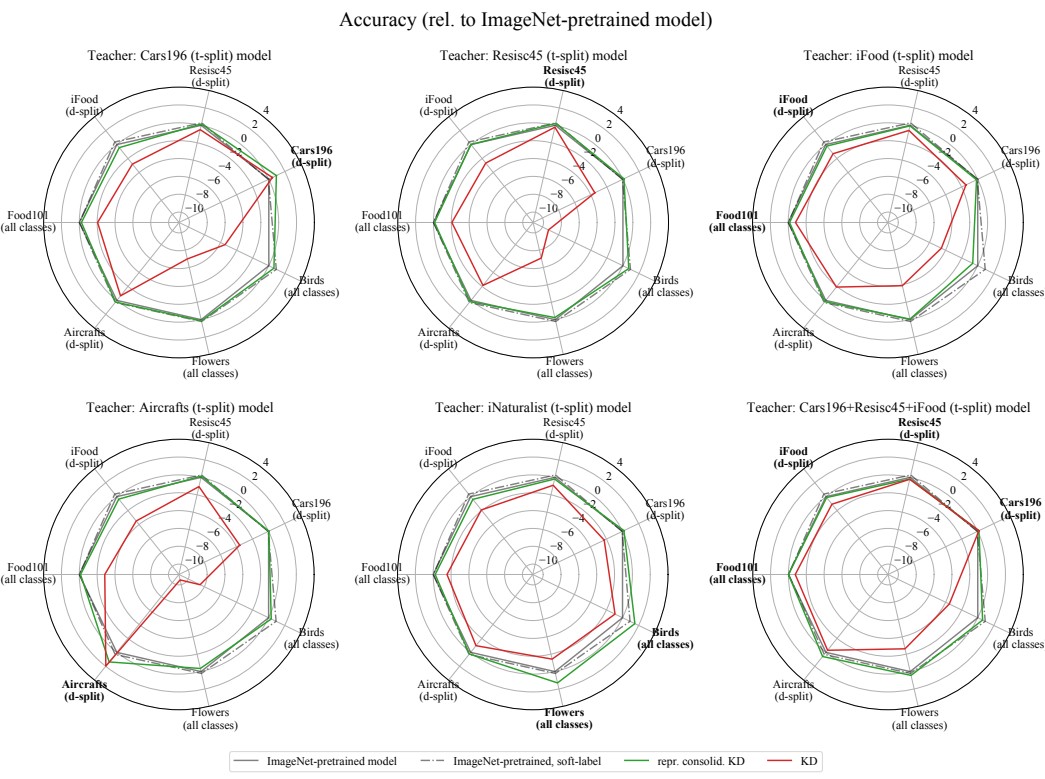

Figure 11: **Exp. 3** $N = 1$ (first 4 graphs) and $N > 1$ (last graph) teacher(s), **full**-shot **fine-tuning** downstream transfer. Part 2/2 of full results for Fig. 4. The same conclusions as the fixed representation scenario in Figs. 2, 6 hold.

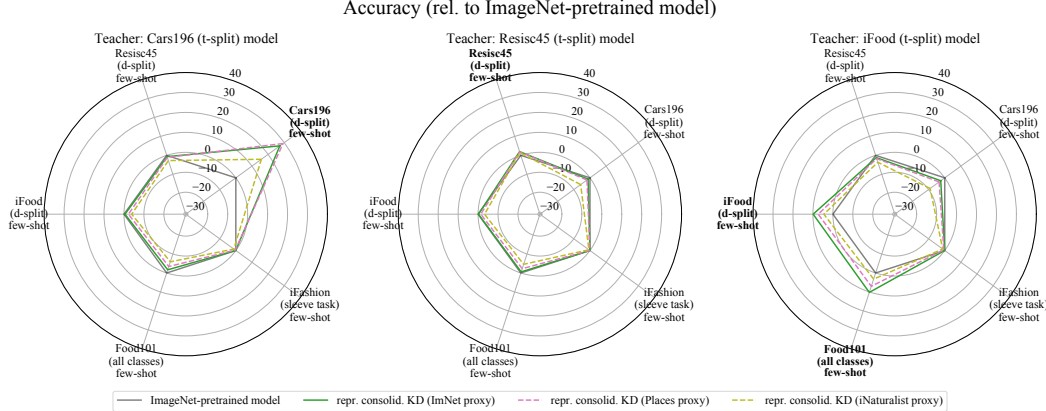

Figure 12: **Exp. 5** $N = 1$, 5-shot linear SVM downstream transfer, different proxies. Full results for Fig. 5b. Conclusions similar – as the proxy, Places365 is similar to ImageNet, but a narrower-scoped iNaturalist underperforms.

**Exp. 3: Fine-tuning downstream.** Figs. 10, 11 show full results for Fig. 4. These extensive results show that in both few-shot and full-shot scenarios, and both single-domain and multi-domain scenarios, we have the same conclusions as the fixed representation experiments.

**Exp. 5: Proxy data choice.** Fig. 12 shows full results for Fig. 5b. This shows similar results with different teacher $\mathcal{D}_t$, *i.e.* Places365 performs similarly to ImageNet as the proxy, but the narrower-scope iNaturalist underperforms.

## C  RAW NUMBER ACCURACY TABLES FOR ALL EXPERIMENTS

- Table 4: fixed representation few-shot results for Figs. 2, 3 (Figs. 6,7,8,9 in the appendix).
- Table 5: fixed representation few-shot results for Fig. 5.
- Table 6: fine-tuning few-shot results for Fig. 10 in the appendix.
- Table 7: fine-tuning full-shot results for Fig. 4 (Figs. 11 in the appendix.)

## D  LIMITATIONS

We assume we have perfect knowledge of which tasks form the same domain and which tasks do not belong to a domain. Our performance drops when teachers from different domains are consolidated. In future work, we plan to automatically determine how to cluster a large amount of teachers into domains. In addition, one of our contributions assume the existence of a strong representation baseline such as the ImageNet pre-trained model, which is true for fields like images and language, but not for others (*e.g.*, 3D reconstruction). Our method also takes the teachers as is and learn from them, and any mistakes made by the teachers can be propagated during the consolidation. Possible mitigations include using a better teacher that makes less such mistakes, or using regularization on both teacher and student training, such as making similar inputs map to similar outputs.

| | Cars196 | Resisc45 | iFood | Food101 (all) | iFashion (sleeve) | DTD | Flowers | Caltech256 | Birds | Aircrafts |
|---|---|---|---|---|---|---|---|---|---|---|
| ImageNet-pretrained model | 31.94 | 70.66 | 28.94 | 36.95 | 89.06 | 60.98 | 83.84 | 80.28 | 54.73 | 30.71 |
| ImageNet-pretrained, soft-label | 31.47 | 69.99 | 29.21 | 37.29 | 89.18 | 60.44 | 82.86 | 81.35 | 57.19 | 29.77 |
| Cars196 (t-split) KD | 59.39 | 51.56 | 9.75 | 11.54 | 84.27 | 36.07 | 59.91 | 35.51 | 16.90 | 16.82 |
| Cars196 (t-split) repr. consolid. KD | 59.21 | 70.00 | 28.42 | 35.28 | 89.00 | 58.99 | 83.01 | 80.26 | 54.02 | 29.85 |
| Cars196 (t-split) CRD | 62.42 | 55.96 | 12.42 | 14.39 | 85.22 | 39.54 | 66.76 | 41.09 | 20.33 | 19.13 |
| Cars196 (t-split) repr. consolid. CRD | 60.97 | 71.02 | 29.14 | 36.15 | 89.10 | 59.47 | 84.33 | 80.23 | 54.09 | 32.04 |
| Cars196 (t-split) MTL oracle | 60.49 | 69.23 | 28.44 | 35.15 | 89.20 | 60.46 | 82.79 | 79.79 | 55.37 | 30.89 |
| Resisc45 (t-split) KD | 8.97 | 61.62 | 9.85 | 12.11 | 84.97 | 37.80 | 56.06 | 34.14 | 11.23 | 11.57 |
| Resisc45 (t-split) repr. consolid. KD | 30.80 | 72.64 | 28.77 | 36.37 | 89.03 | 60.08 | 82.75 | 80.77 | 55.47 | 29.17 |
| Resisc45 (t-split) CRD | 16.81 | 68.62 | 16.95 | 19.84 | 86.42 | 45.00 | 72.36 | 49.02 | 20.89 | 18.31 |
| Resisc45 (t-split) repr. consolid. CRD | 32.66 | 74.03 | 29.61 | 36.99 | 89.10 | 60.46 | 84.27 | 80.73 | 55.19 | 31.08 |
| Resisc45 (t-split) MTL oracle | 32.51 | 73.06 | 29.07 | 36.21 | 89.11 | 59.82 | 83.74 | 80.04 | 57.11 | 31.19 |
| iFood (t-split) KD | 12.85 | 53.57 | 35.26 | 44.05 | 85.39 | 36.62 | 70.99 | 39.94 | 17.95 | 14.90 |
| iFood (t-split) repr. consolid. KD | 29.44 | 69.49 | 38.85 | 47.19 | 89.04 | 57.16 | 82.66 | 79.17 | 51.07 | 27.31 |
| iFood (t-split) CRD | 14.80 | 56.57 | 36.30 | 44.67 | 86.03 | 38.95 | 74.03 | 42.56 | 19.91 | 16.18 |
| iFood (t-split) repr. consolid. CRD | 30.54 | 70.51 | 39.52 | 47.58 | 88.89 | 57.48 | 84.02 | 78.67 | 50.35 | 28.57 |
| iFood (t-split) MTL oracle | 32.57 | 69.41 | 38.88 | 46.65 | 89.13 | 59.46 | 82.82 | 79.52 | 56.37 | 31.50 |
| iFashion (category task) KD | 6.41 | 37.97 | 6.26 | 8.09 | 90.70 | 23.45 | 44.01 | 30.32 | 9.28 | 9.27 |
| iFashion (category task) repr. consolid. KD | 31.89 | 70.02 | 29.23 | 37.45 | 89.15 | 60.76 | 83.28 | 81.29 | 57.10 | 30.24 |
| iFashion (category task) CRD | 16.11 | 57.91 | 14.74 | 18.35 | 90.93 | 41.77 | 72.11 | 50.08 | 20.87 | 17.52 |
| iFashion (category task) repr. consolid. CRD | 32.36 | 71.75 | 29.91 | 37.60 | 89.72 | 60.98 | 84.65 | 80.91 | 55.51 | 31.39 |
| iFashion (category task) MTL oracle | 31.91 | 69.35 | 28.78 | 36.27 | 88.75 | 60.50 | 83.40 | 79.93 | 57.01 | 31.28 |
| DTD (t-split) KD | 11.29 | 53.12 | 14.11 | 19.00 | 86.39 | 46.80 | 62.48 | 46.38 | 17.14 | 14.75 |
| DTD (t-split) repr. consolid. KD | 31.33 | 69.78 | 29.17 | 37.38 | 89.08 | 61.43 | 82.80 | 81.28 | 56.09 | 29.59 |
| DTD (t-split) CRD | 23.39 | 64.77 | 24.59 | 31.08 | 87.82 | 55.80 | 79.29 | 66.36 | 33.82 | 24.15 |
| DTD (t-split) repr. consolid. CRD | 33.10 | 71.58 | 29.93 | 38.27 | 89.10 | 61.47 | 84.96 | 81.08 | 56.02 | 31.35 |
| DTD (t-split) MTL oracle | 32.76 | 69.56 | 29.24 | 36.95 | 89.24 | 61.10 | 83.22 | 80.57 | 57.66 | 31.05 |
| Flowers (t-split) KD | 16.72 | 58.08 | 16.68 | 20.67 | 87.03 | 44.70 | 77.35 | 54.49 | 31.32 | 20.26 |
| Flowers (t-split) repr. consolid. KD | 31.81 | 69.94 | 29.46 | 37.01 | 89.13 | 60.06 | 84.79 | 81.18 | 57.07 | 30.41 |
| Flowers (t-split) CRD | 26.84 | 64.86 | 22.78 | 27.54 | 88.18 | 50.05 | 82.66 | 66.27 | 42.05 | 27.22 |
| Flowers (t-split) repr. consolid. CRD | 33.79 | 71.38 | 30.05 | 37.68 | 89.25 | 60.42 | 86.37 | 81.10 | 56.56 | 32.52 |
| Flowers (t-split) MTL oracle | 32.15 | 68.89 | 27.85 | 35.19 | 89.11 | 58.78 | 82.73 | 79.72 | 55.97 | 30.79 |
| Caltech256 (t-split) KD | 21.65 | 59.67 | 19.44 | 25.10 | 87.35 | 51.59 | 75.52 | 68.98 | 36.09 | 23.32 |
| Caltech256 (t-split) repr. consolid. KD | 31.71 | 69.36 | 29.18 | 36.97 | 89.08 | 61.07 | 83.14 | 81.28 | 56.63 | 29.98 |
| Caltech256 (t-split) CRD | 27.63 | 63.52 | 23.77 | 30.08 | 88.25 | 55.35 | 80.37 | 73.42 | 43.88 | 27.22 |
| Caltech256 (t-split) repr. consolid. CRD | 33.44 | 70.98 | 30.19 | 37.89 | 89.27 | 61.20 | 84.70 | 81.22 | 56.68 | 31.61 |
| Caltech256 (t-split) MTL oracle | 30.61 | 67.73 | 27.64 | 34.92 | 89.09 | 58.92 | 83.04 | 78.83 | 56.40 | 29.21 |
| Birds (t-split) KD | 17.15 | 57.95 | 17.06 | 20.72 | 87.03 | 46.37 | 72.16 | 53.26 | 62.41 | 19.00 |
| Birds (t-split) repr. consolid. KD | 31.50 | 70.35 | 29.16 | 36.83 | 89.20 | 60.32 | 83.30 | 80.89 | 66.37 | 29.65 |
| Birds (t-split) CRD | 23.22 | 62.63 | 21.31 | 25.92 | 88.10 | 50.80 | 77.85 | 61.92 | 64.00 | 23.10 |
| Birds (t-split) repr. consolid. CRD | 32.70 | 71.65 | 29.72 | 37.52 | 89.23 | 60.38 | 84.63 | 80.56 | 66.35 | 31.54 |
| Birds (t-split) MTL oracle | 31.89 | 69.85 | 29.12 | 36.54 | 89.30 | 60.11 | 83.00 | 80.25 | 64.79 | 30.26 |
| Aircrafts (t-split) KD | 10.56 | 44.30 | 7.81 | 9.09 | 84.17 | 29.16 | 48.32 | 29.18 | 10.87 | 58.51 |
| Aircrafts (t-split) repr. consolid. KD | 30.71 | 69.58 | 28.25 | 35.54 | 89.06 | 58.67 | 81.36 | 80.54 | 53.95 | 55.26 |
| Aircrafts (t-split) CRD | 14.06 | 50.09 | 10.56 | 11.86 | 85.17 | 33.13 | 57.85 | 35.16 | 14.11 | 60.44 |
| Aircrafts (t-split) repr. consolid. CRD | 32.54 | 70.84 | 28.91 | 36.15 | 89.17 | 59.31 | 83.01 | 80.27 | 53.29 | 57.25 |
| Aircrafts (t-split) MTL oracle | 31.55 | 69.92 | 28.91 | 36.30 | 89.28 | 60.40 | 83.04 | 79.65 | 56.31 | 54.82 |
| iNaturalist (t-split) KD | 13.51 | 57.25 | 14.66 | 18.68 | 84.46 | 39.49 | 79.74 | 41.59 | 57.49 | 16.98 |
| iNaturalist (t-split) repr. consolid. KD | 30.08 | 69.67 | 27.35 | 34.52 | 88.70 | 57.28 | 84.77 | 78.01 | 67.58 | 27.94 |
| iNaturalist (t-split) CRD | 14.92 | 58.76 | 15.30 | 19.41 | 84.84 | 40.42 | 80.98 | 42.71 | 58.48 | 18.00 |
| iNaturalist (t-split) repr. consolid. CRD | 30.87 | 70.34 | 27.25 | 33.97 | 88.81 | 56.53 | 86.06 | 77.08 | 67.45 | 28.93 |
| iNaturalist (t-split) MTL oracle | 32.14 | 69.37 | 28.45 | 35.00 | 89.25 | 58.80 | 85.42 | 79.05 | 68.53 | 29.93 |
| Cars196 10% classes x 5 (Res18) repr. consolid. CRD | 44.04 | 71.89 | 30.45 | 37.80 | 89.20 | 60.56 | 85.60 | 81.01 | 58.09 | 32.86 |
| Cars196 10% classes x 5 (Res18) repr. consolid. KD | 42.03 | 69.95 | 29.44 | 36.97 | 89.05 | 60.15 | 83.23 | 81.48 | 58.18 | 31.10 |
| Cars196 10% classes x 5 (Res18) CRD | 48.45 | 66.00 | 23.35 | 27.10 | 88.42 | 52.48 | 80.29 | 66.12 | 47.90 | 29.24 |
| Cars196 10% classes x 5 (Res18) KD | 45.37 | 60.74 | 17.53 | 20.80 | 87.11 | 48.16 | 73.66 | 58.56 | 39.02 | 23.85 |
| Cars196 10% classes (Res18) repr. consolid. CRD | 39.56 | 72.50 | 29.86 | 37.12 | 89.36 | 60.73 | 85.07 | 80.00 | 57.67 | 33.79 |
| Cars196 10% classes (Res18) repr. consolid. KD | 36.97 | 69.64 | 28.62 | 35.95 | 88.92 | 59.88 | 82.79 | 80.37 | 56.80 | 30.87 |
| Cars196 10% classes (Res18) teacher (best of 5) | 35.39 | 56.47 | 16.98 | 20.50 | 87.59 | 45.77 | 70.58 | 60.41 | 41.87 | 23.40 |
| iFood 10% classes x 5 (Res18) repr. consolid. CRD | 32.36 | 71.87 | 35.93 | 42.38 | 89.24 | 58.56 | 84.70 | 79.37 | 52.72 | 31.52 |
| iFood 10% classes x 5 (Res18) repr. consolid. KD | 31.14 | 70.69 | 35.82 | 42.97 | 89.08 | 58.98 | 83.51 | 80.70 | 53.91 | 29.80 |
| iFood 10% classes x 5 (Res18) CRD | 22.26 | 64.98 | 34.87 | 39.92 | 87.67 | 48.94 | 79.60 | 58.12 | 33.24 | 24.30 |
| iFood 10% classes x 5 (Res18) KD | 17.60 | 60.26 | 34.10 | 39.71 | 86.58 | 45.93 | 76.18 | 51.55 | 27.91 | 20.27 |
| iFood 10% classes (Res18) repr. consolid. CRD | 32.75 | 71.36 | 32.22 | 38.97 | 89.29 | 59.18 | 84.45 | 78.88 | 54.07 | 32.17 |
| iFood 10% classes (Res18) repr. consolid. KD | 30.45 | 69.54 | 31.25 | 38.47 | 88.99 | 58.59 | 81.97 | 79.37 | 54.33 | 30.25 |
| iFood 10% classes (Res18) teacher (best of 5) | 14.27 | 52.35 | 23.11 | 26.40 | 86.45 | 38.34 | 65.88 | 44.94 | 22.58 | 16.11 |
| Cars196 10% classes x 5 repr. consolid. CRD | 46.69 | 71.65 | 30.10 | 37.72 | 89.09 | 60.48 | 85.50 | 80.75 | 57.01 | 33.00 |
| Cars196 10% classes x 5 repr. consolid. KD | 45.99 | 70.10 | 29.23 | 37.02 | 89.16 | 60.40 | 83.82 | 81.21 | 57.40 | 31.42 |
| Cars196 10% classes x 5 CRD | 53.22 | 67.42 | 23.59 | 28.19 | 88.17 | 54.09 | 81.42 | 66.77 | 45.41 | 29.87 |
| Cars196 10% classes x 5 KD | 50.33 | 62.63 | 18.52 | 22.32 | 86.77 | 49.31 | 75.66 | 59.74 | 37.27 | 25.48 |
| Cars196 10% classes repr. consolid. CRD | 41.67 | 71.77 | 29.77 | 37.10 | 89.32 | 60.52 | 85.37 | 80.05 | 56.80 | 33.08 |
| Cars196 10% classes repr. consolid. KD | 39.23 | 69.62 | 28.42 | 36.01 | 89.03 | 59.84 | 82.76 | 80.34 | 56.43 | 31.20 |
| Cars196 10% classes teacher (best of 5) | 42.92 | 64.48 | 20.87 | 25.95 | 87.58 | 51.79 | 78.69 | 64.35 | 43.20 | 25.01 |
| iFood 10% classes x 5 repr. consolid. CRD | 32.12 | 72.04 | 37.14 | 44.30 | 89.09 | 58.97 | 85.69 | 79.26 | 52.23 | 31.47 |
| iFood 10% classes x 5 repr. consolid. KD | 30.74 | 70.64 | 37.34 | 44.77 | 88.91 | 59.17 | 84.58 | 80.20 | 52.47 | 29.83 |
| iFood 10% classes x 5 CRD | 21.95 | 65.63 | 37.04 | 43.06 | 87.12 | 49.35 | 81.66 | 56.65 | 30.24 | 24.08 |
| iFood 10% classes x 5 KD | 17.70 | 61.33 | 35.88 | 42.71 | 86.18 | 44.63 | 77.50 | 50.98 | 25.36 | 20.49 |
| iFood 10% classes repr. consolid. CRD | 32.64 | 70.85 | 32.81 | 39.57 | 89.19 | 58.40 | 84.36 | 79.19 | 53.73 | 31.94 |
| iFood 10% classes repr. consolid. KD | 30.05 | 68.62 | 32.06 | 39.37 | 88.83 | 58.67 | 82.09 | 79.43 | 53.83 | 29.68 |
| iFood 10% classes teacher (best of 5) | 13.85 | 54.31 | 25.16 | 29.22 | 86.48 | 39.39 | 69.26 | 44.21 | 20.54 | 16.08 |
| Cars196+Resisc45+iFood (t-split) repr. consolid. KD | 47.85 | 72.39 | 34.57 | 41.92 | 89.09 | 58.63 | 83.64 | 79.90 | 51.79 | 29.18 |
| Cars196+Resisc45+iFood (t-split) KD | 53.27 | 66.27 | 34.44 | 41.84 | 85.32 | 43.74 | 75.84 | 48.16 | 23.16 | 19.99 |
| Cars196 + Resisc45 + iFood (t-split) repr. consolid. (Places+iNat+logo2k+ImNet proxy 10 epochs) | 45.71 | 73.00 | 32.87 | 39.80 | 89.00 | 59.09 | 83.89 | 79.83 | 52.40 | 29.46 |
| Cars196 + Resisc45 + iFood (t-split) trad. distill (Places+iNat+logo2k+ImNet proxy 10 epochs) | 53.26 | 66.34 | 32.80 | 39.10 | 85.43 | 43.81 | 75.92 | 47.75 | 23.15 | 20.15 |

Table 4: Fixed representation + linear SVM, 5-shot results (Part 1/2) raw numbers for Figs. 2, 3 in the main paper (Figs. 6, 7, 8, 9 in the appendix)

| | Cars196 | Resisc45 | iFood | Food101 (all) | iFashion (sleeve) |
|---|---|---|---|---|---|
| Cars196 (t-split) repr. consolid. KD old:new = 1:3 | 61.21 | 67.83 | 26.70 | 33.35 | 88.92 |
| Cars196 (t-split) repr. consolid. KD old:new = 1:1 | 59.21 | 70.00 | 28.42 | 35.28 | 89.00 |
| Cars196 (t-split) repr. consolid. KD old:new = 3:1 | 52.80 | 70.82 | 29.18 | 36.53 | 89.13 |
| Cars196 (t-split) repr. consolid. KD (ImNet proxy) | 59.21 | 70.00 | 28.42 | 35.28 | 89.00 |
| Cars196 (t-split) repr. consolid. KD (Places proxy) | 61.18 | 69.59 | 26.69 | 33.54 | 88.68 |
| Cars196 (t-split) repr. consolid. KD (Places proxy + label) | 53.16 | 69.32 | 23.17 | 28.54 | 88.23 |
| Cars196 (t-split) KD (Places proxy) | 59.98 | 51.11 | 9.54 | 11.10 | 84.38 |
| Cars196 (t-split) repr. consolid. KD (iNaturalist proxy) | 47.76 | 67.90 | 25.08 | 31.07 | 87.80 |
| Resisc45 (t-split) repr. consolid. KD (ImNet proxy) | 30.80 | 72.64 | 28.77 | 36.37 | 89.03 |
| Resisc45 (t-split) repr. consolid. KD (Places proxy) | 30.08 | 72.56 | 27.59 | 34.68 | 88.99 |
| Resisc45 (t-split) repr. consolid. KD (iNaturalist proxy) | 26.17 | 72.11 | 25.71 | 32.38 | 88.40 |
| iFood (t-split) repr. consolid. KD (ImNet proxy) | 29.44 | 69.49 | 38.85 | 47.19 | 89.04 |
| iFood (t-split) repr. consolid. KD (Places proxy) | 28.43 | 68.94 | 36.30 | 43.97 | 88.50 |
| iFood (t-split) repr. consolid. KD (iNaturalist proxy) | 22.87 | 67.04 | 33.47 | 40.18 | 87.79 |

Table 5: Fixed representation + linear SVM, 5-shot results (Part 2/2) raw numbers for Fig. 5 in the main paper.

| | Cars196 | Resisc45 | iFood | Food101 (all) | iFashion (sleeve) |
|---|---|---|---|---|---|
| ImageNet-pretrained model | 37.18 | 68.91 | 28.25 | 34.72 | 89.68 |
| ImageNet-pretrained, soft-label | 39.84 | 69.88 | 28.72 | 35.31 | 89.92 |
| Cars196 (t-split) repr. consolid. | 55.80 | 69.91 | 28.41 | 34.00 | 89.91 |
| Cars196 (t-split) trad. distill | 60.19 | 61.30 | 18.98 | 19.79 | 87.90 |
| Resisc45 (t-split) repr. consolid. | 38.30 | 71.71 | 28.16 | 34.33 | 89.83 |
| Resisc45 (t-split) trad. distill | 21.12 | 60.93 | 17.57 | 19.46 | 88.05 |
| iFood (t-split) repr. consolid. | 35.94 | 68.39 | 35.27 | 42.14 | 89.54 |
| iFood (t-split) trad. distill | 27.89 | 61.58 | 31.23 | 39.33 | 87.91 |
| iFashion category task repr. consolid. | 38.57 | 68.45 | 28.80 | 35.88 | 89.29 |
| iFashion category task trad. distill | 16.71 | 54.53 | 11.49 | 14.93 | 89.80 |
| Cars196 + Resisc45 + iFood (t-split) repr. consolid. | 45.31 | 70.84 | 30.90 | 37.62 | 89.87 |
| Cars196 + Resisc45 + iFood (t-split) trad. distill | 45.29 | 67.05 | 30.14 | 35.44 | 87.97 |

Table 6: Fine-tuning, 5-shot results raw numbers for Fig. 10 in the appendix

| | Cars196 | Resisc45 | iFood | Food101 (all) | Aircrafts | Flowers (all) | Birds (all) |
|---|---|---|---|---|---|---|---|
| ImageNet-pretrained model | 90.90 | 96.93 | 78.55 | 88.15 | 87.57 | 92.32 | 80.82 |
| ImageNet-pretrained, soft-label | 90.85 | 97.20 | 78.92 | 88.13 | 87.87 | 92.54 | 81.76 |
| Cars196 (t-split) repr. consolid. | 91.84 | 97.08 | 78.12 | 87.91 | 87.81 | 92.45 | 81.46 |
| Cars196 (t-split) trad. distill | 91.35 | 96.43 | 75.76 | 86.10 | 86.91 | 85.35 | 75.39 |
| Resisc45 (t-split) repr. consolid. | 91.03 | 97.14 | 78.57 | 88.02 | 87.75 | 92.08 | 81.55 |
| Resisc45 (t-split) trad. distill | 87.41 | 96.71 | 75.90 | 86.09 | 85.42 | 85.25 | 71.57 |
| iFood (t-split) repr. consolid. | 90.75 | 96.86 | 78.34 | 88.00 | 87.75 | 92.26 | 80.20 |
| iFood (t-split) trad. distill | 89.44 | 96.34 | 77.24 | 87.33 | 85.65 | 88.42 | 76.30 |
| iFood (t-split) teacher | 89.09 | 96.21 | 77.76 | 87.92 | 84.46 | 87.97 | 75.92 |
| Aircrafts (t-split) repr. consolid. | 90.93 | 97.08 | 78.20 | 88.01 | 88.94 | 91.97 | 81.15 |
| Aircrafts (t-split) trad. distill | 87.31 | 95.87 | 75.06 | 85.29 | 89.54 | 81.79 | 72.30 |
| iNaturalist (t-split) repr. consolid. | 91.08 | 96.74 | 78.16 | 87.92 | 87.81 | 93.62 | 82.36 |
| iNaturalist (t-split) trad. distill | 88.62 | 96.02 | 76.64 | 86.60 | 86.61 | 90.89 | 79.89 |
| iNaturalist (t-split) teacher | 89.37 | 96.37 | 76.89 | 87.09 | 85.59 | 91.22 | 79.62 |
| Cars196 + Resisc45 + iFood (t-split) repr. consolid. | 91.05 | 96.74 | 78.42 | 88.15 | 88.16 | 92.76 | 81.43 |
| Cars196 + Resisc45 + iFood (t-split) trad. distill | 91.05 | 96.71 | 77.47 | 87.38 | 87.27 | 89.71 | 77.29 |

Table 7: Fine-tuning, full-shot results raw numbers for Fig. 4 in the main paper (Figs. 11 in the appendix.

