# OpenReview forum: "Representation Consolidation from Multiple Expert Teachers"
_ICLR.cc/2022/Conference — ICLR 2022 Submitted_

### Official Review · Reviewer_HSTC · 2021-11-01

**Correctness:** 3
**Technical Novelty And Significance:** 2
**Empirical Novelty And Significance:** 3
**Recommendation:** 5
**Confidence:** 4

**Main Review:**

Strengths:

1. This paper is well written and easy to follow. I think the study multi-teacher consolidation problem is interesting and important.  In some practical domains (like healthcare), it is difficult to construct a large dataset to train a model,  how to conduct transfer learning from multiple datasets (models) is critical.

2. The authors conduct extensive analysis on multiple datasets.

Weakness:
1. The technical contribution is limited, as the method is somehow ad-hoc. Different from previous multi-teacher-distillation works, the authors add a generalized model (ImageNet model) during knowledge distillation to prevent representation collapse. However, there is no deep analysis of why this can achieve such a goal. Moreover, this strategy simply treats the generalized model and other teacher models equally, while there should be different roles during the distillation between the generalized model and teacher models.

2. As the technical contribution is limited, I would appreciate the merits of experiments.  Although the authors conducted extensive analysis, most of them are in a few-shot learning setting (i.e., using only several images to train the downstream networks) not a fine-tuning setting. For the few-shot learning setting, the performance of different experimental runs has large perturbations, it is truly difficult to determine the effectiveness of the proposed method.

3. The authors also conduct the experiments under full fine-tuning in Exp. 3. However, the improvement of the proposed method compared with "imagenet-pretrained + fine-tuning" is quite incremental (see Figure 4). Therefore, it is difficult to sufficiently support the advantage of the proposed multi-teacher consideration method with the existing experiments.

Other comments:

1. In the first paragraph, the authors mention the overall goal is to "enrich the diversity of the expert library by accumulating knowledge from transferred downstream tasks" and in the second paragraph, the authors argue that adding fine-tuned downstream models back into the library is naive and does not scale. However, to achieve the overall goal, the proposed multi-model consideration method still seems to depend on this naive strategy. Maybe the authors can reorganize the motivation part in the introduction to avoid confusion.

2.  It is better if the authors can briefly summarize the size/class distribution of each dataset in the experimental setup section.

3.  As there are performance perturbations for different experimental runs, especially under the few-shot learning setting. Are the results in the tables the average values of different runs or single runs? If they are the results of different runs, it is better to show the std of the results.

**Summary Of The Paper:**

This paper studies the problem of multi-model consolidation, which aims to merge (consolidate) the knowledge from multiple models into one model so that we can better transfer this merged model for downstream tasks. Different from the traditional knowledge distillation or multi-teacher distillation, the proposed method aims to transferability of merged models for the downstream tasks, whilst achieving good performance on source tasks. The key technique (technical contribution) is to add a generalized model during the knowledge distillation, which can be regarded as s "special" teacher model.

The authors conduct extensive experiments on multiple datasets and most of the experiments are under a few-shot linear probe transfer learning setting.

**Summary Of The Review:**

The studied problem is interesting and important and the whole paper is easy to follow, while the technical contribution is limited and the experiments are not very strong to support the advantage of the proposed method.

---

> ### Author Response · Authors · 2021-11-20
> **Response for Reviewer HSTC**
>
> We thank the reviewer for the thorough review and thoughtful feedback! We address the following concerns:
>
> Analysis on why adding the generalist performs well
>
> * We do provide a detailed analysis and guide our readers through our thought process in Section 2 “Forgetting and representation collapse during distillation”, based on the results in Section 4 Exp. 0 and Table 2.
> * We show that the assumption that “because the existing networks work well for the upstream task $D_t^1$, distilling it will transfer well to the downstream task $D_d^j$” is ill-informed, since it actually underperforms. (3rd row, “KD (trad. distill)”)
> * We further show that the teacher itself underperforms when transferred to a related downstream task (2nd row, “Teacher (t-split)”). It naturally follows that a model distilled only on these teachers will not work well.
>     * We then notice that these teachers are initialized with ImageNet pre-train (1st row), which works well on downstream tasks, and only after fine-tuning they underperform. It naturally follows that the fine-tuning on the teacher’s task causes the performance drop.
> * We conclude that the forgetting in teacher’s training (fine-tuning) causes the performance degradation. It naturally follows that adding the generalist teacher, which the teachers were initialized with, will mitigate the forgetting brought by the teachers.
> * If necessary, we can reiterate the thought process in Section 4 as we present the results as well.
>
> Few-shot experiments
>
> * Detailed in our experiment setup, we average the few-shot performance over 50 runs (different subsamples) since they are faster to run. The extensive experiments’ observations hold for a large set of upstream/downstream tasks and across few-shot/full-shot and fixed-backbone/fine-tuning.
>
> Fine-tuning doesn’t work?
>
> * On the contrary, we demonstrate that our methods works very well on fine-tuning in Figure 10 (appendix), e.g. +18% for ImageNet+Cars196→Cars196, and +7% for ImageNet+iFood→Food101. The performance is limited only in the *full-shot* fine-tuning scenario (Figure 11), although we still reliably outperform traditional distillation. The difference is how much data the downstream task has (which is outside the method’s control), not fine-tuning vs. fixed representation.
> * In general, pre-training and representation learning work best when the downstream task is data-deprived. When the downstream task has sufficient data, pre-training is not as useful. [a] This is a valid limitation of our field, and many papers only focus on few-shot for downstream tasks. Note that Figure 10 tries to leverage knowledge learned from mid-sized datasets such as Cars196, and transfers downstream to a similarly mid-sized dataset split, so it is understandable that the improvement is small.
>     * [a] He, Kaiming, Ross Girshick, and Piotr Dollár. "Rethinking imagenet pre-training." ICCV 2019.
> * If reviewers consider it useful, we could swap the few-shot fine-tuning experiments into the main text and put full-shot fine-tuning into the appendix to better highlight this argument.
>
> Treating the generalized model and other teacher models equally?
>
> * We put more significance in maintaining the generalist’s knowledge than other teachers, motivated by the importance of maintaining the pre-trained representational power. In eq. 1, we set $\lambda_0$ to 1 and $\lambda_1$ to 1/N where N is the number of task-specific teachers. This means that even though there might be N=5 teachers in Fig. 3, the generalist is still weighted 1 and the others 0.2.
> * For future work, it is indeed possible to perform other incremental learning regularizations such as EWC or SI from the incremental learning literature, but it is out of the scope of this paper which aims to provide one solution to the forgetting issue we identify.
>
> Clarify whether we add everything to the expert library?
>
> * Thank you for the suggestion! We will bring up in the revision that our method can be used to shrink the expert library by a significant factor, by merging multiple related representations into one.
>
> Size/class distribution of each dataset
>
> * Great point! We will summarize this into a table in the appendix, and add a link to it in the main paper.

---

> > ### Comment · Reviewer_HSTC · 2021-11-29
> > **Thanks for the response**
> >
> > Thanks to the authors for the response. After reading other reviewers' comments and responses, I will maintain the original score.

---

### Official Review · Reviewer_Swfw · 2021-11-02

**Correctness:** 3
**Technical Novelty And Significance:** 2
**Empirical Novelty And Significance:** 3
**Recommendation:** 3
**Confidence:** 4

**Main Review:**

Strengths:

1. The problem of how to properly use multiple pre-trained models for downstream transfer learning is interesting and practical in real life.

2. The introduced method achieves better performance than distillation from a single ImageNet-pretrained teacher or task-specfic teacher.

Weaknesses:

1. This paper tells a big story of ``life-long meta-learning'' in the beginning, but fails to verify it in the method and experiment part. Specifically, the authors argue that they would like to automatically enrich the diversity of the pre-trained model zoo by accumulating knowledge from downstream tasks. However, I cannot find any evidence of enriching the diversity in this paper.

2. The authors claim that the generalist teacher is important to the transferability of distilled representations. It should be important to carefully discuss the choice of the generalist teacher in this framework, e.g., how does it perform when adopting different or multiple generalist teachers. I did not find anything about it.

3. Similarly, how to properly select task-specific teachers for corresponding downstream tasks? How much does it affect when choosing different numbers of task-specific teachers? In the experimental part, the authors only show the comparisons when adopting ad-hoc teachers, which is less convincing.

4. As shown in Fig. 1, I find the comparison between *distillation directly to the downstream task* and *distillation first and then finetuning to the downstream task* interesting. It is necessary to conduct ablation studies on this comparison while keeping the teachers (a generalist teacher + multiple task-specific teachers) the same.


====================

Post-rebuttal:

Thanks for the authors' efforts. But unfortunately, I choose to maintain my original score considering the other reviewers' concerns and the overclaiming problem (the method and experiments fail to support the motivation) as I mentioned in the initial review.

**Summary Of The Paper:**

The paper introduces a representation consolidation method to properly aggregate the pre-trained knowledge from multiple teachers for transfer learning. It claims that a generalist teacher is necessary for preserving the transferability of distilled representations, and task-specific teachers contribute to better performance in the same-domain downstream tasks. The introduced method performs on par with multi-task training while neglecting the need for teacher datasets.

**Summary Of The Review:**

The motivation of this paper (life-long meta-learning) is attractive, but the method and experiments fail to well support it. I tend to reject this paper, and recommend the authors refine it and submit it to the next conference.

---

> ### Author Response · Authors · 2021-11-20
> **Response for Reviewer Swfw**
>
> We thank the reviewer for the time for the review and valuable feedback. We address the concerns raised in the review as follows:
>
> Clarification regarding life-long learning and our goal
>
> * We only introduce the topic of life-long learning as the long-term motivation of our work. Although it is our ultimate goal, the main goal of our paper is to merge multiple representations into one, which is a necessary building block to life-long learning based on an expert library. We list the main points of our work at the end of the introduction section, which do not claim to solve expert-library-based life-long learning. Other building blocks that are needed are also under-explored in the literature, and we would love to work on them in the future. We will fix the introduction to avoid this confusion.
>
> Exploring generalist teachers
>
> * We agree that the choice of the generalist teacher is important, and we do have a well-motivated guideline of doing so (choose the pre-trained network that the teachers started from and is already known to work well for the downstream tasks). The choice of generalist will be an interesting future direction to explore, e.g. using both ImageNet and Places365 pre-trained models as the generalists.
>
> Selecting task-specific teachers for downstream tasks
>
> * Predicting the best teacher to adapt from *given downstream task data* exists in the literature (see paper citations Puigcerver et al. (2020); Achille et al. (2019); Deshpande et al. (2021)). We expect that they are compatible with our set of consolidated models, since our findings (data similar to the downstream task, especially ones split from the same dataset, best boost performance) are in line with the assumptions of these methods.
>     * In our experiments, we do not choose the teachers ad-hoc, but rather demonstrate the performance transferring from all possible representations in the expert library, as shown in Figure 6 (appendix) and summarized in Fig. 2b (main text). This can be seen as an oracle to the specific model selection method that one may use.
> * But in practice, at the time of model merging, we cannot forecast which specific set of downstream tasks a representation will be used on in the future. We do not propose merging models on-the-fly given a specific downstream task, since it will be computationally very inefficient (than choosing from a short curated list of merged models). We can only broaden the diversity of the expert library in general.
> * We do analyze our results for distilling multiple teachers with respect to downstream tasks, in Figures 3a and 3b.
>     * When the multiple teachers come from the same domain (Fig. 3a), consolidating 5 teachers works better than consolidating only 1 teacher for the related downstream task, but performs similarly for unrelated tasks.
>     * When the multiple teachers come from different domains (Fig. 3b), consolidating them only outperforms the ImageNet baseline, and is worse on the respective domains than consolidating just one teacher on that domain.
>
> Fig. 1 distilling directly to the downstream task?
>
> * It is inherently impossible to distill directly to the downstream task, because the downstream task is novel (it consists of classes not seen in any upstream model). The main message of Fig. 1 is that the downstream task is *different* from the task of the teachers when it comes to using distillation for downstream transfer learning. It is only possible to distill directly to the *original* tasks of the teachers, which is not transfer learning (but we do have sanity-check experiments on those in Table 2, where we show original tasks and novel tasks behave differently). Please let us know if our answer addresses your concern.

---

### Official Review · Reviewer_vU3z · 2021-11-02

**Correctness:** 3
**Technical Novelty And Significance:** 2
**Empirical Novelty And Significance:** 2
**Recommendation:** 5
**Confidence:** 4

**Main Review:**

The main contribution of the paper is the  _representation consolidation_ method to consolidate knowledge from multiple task-specific teacher models into a single expert student, given a generalist teacher is available. This boils down to adding a generalist teacher to the training.

The main findings are that consolidating a task-specific teacher with a generalist teacher is sufficient to rescue the student and consolidation performs only slightly worse/similar to a multi-task joint training oracle, at least on image classification tasks. Which brings about one of the main issues of this paper - despite proposing a general framework, only the image classification problem has been addressed. I would have liked to see more complex tasks such as dense predictions can benefit from the proposed method (e.g.., optical flow, depth estimation). The main insight here is - if we can get a generalist that works well on most scenarios, could we squeeze additional performance from the proposed method or not? Otherwise, the main claim of this paper would be - keeping a generalist teacher improves _any_ transfer learning. Which is kind of a bold statement for ImageNet / iFood testing only, IMHO.

Furthermore, most of the times few-shot learning is illustrated, the few examples of full fine tuning are less favorable to the proposed method.

Otherwise, the paper is well written and illustrated, and the experiments support the claims, in the scenario proposed by the authors.


Strenghts:
- general method for a better distilled student
- easy to implement, provided a generalist is available for the proposed task
- no data replaying for the generalist network during fine-tuning

Weaknessses:
- simple training scenario, mostly classification, few-shot learning
- where fine-tuning is employed (Fig 4/ 10) the results are less convincing

**Summary Of The Paper:**

The paper proposes a task-agnostic Knowledge Distillation method that includes a generalist model as an additional teacher, to limit student forgetting and representation collapse. The authors call this method representation consolidation and claim it can benefit both related and unrelated downstream tasks. The authors experiment with ImageNet and iFood image classification tasks, the accuracy of the proposed method is generally higher than using Knowledge Distillation only.

**Summary Of The Review:**

I lean towards weakly rejecting the paper, based on the limited applicability/experiments mentioned above. If there is a general claim to be made, I would like to see more evidence and other domains tested, beyond image classification. As it stands, I do not find the technical contribution sufficient for acceptance.

---

> ### Author Response · Authors · 2021-11-20
> **Response to Reviewer vU3z**
>
> We thank the reviewer for the thoughtful thinking and thorough review, and we appreciate the feedback!  We address the following concerns:
>
> Generalization to other fields
>
> * We apologize for the confusion. The paper aims to be a classification-only work, and we will happily re-work the language to indicate that the paper’s scope is limited to vision classification tasks. We would like to point out that unfortunately most distillation work focus on classification, and the distillation literature on detection and dense prediction is quite limited. We agree that these would be interesting future directions to explore if our observed phenomenon generalizes.
>
> ImageNet/iFood-only testing?
>
> * We experiment on 9 domains, including Cars196 (car make/models), Resisc45 (satellite imaging), iFood/Food101,  iFashion (clothes), DTD (textures), Flowers, Caltech256 (general objects), Birds, Aircrafts.
> * Our main paper includes results transferring *from* Resisc45, iFood, and Cars196, but we test results transferring to *all 9* domains listed above. Results transferring from other domains are detailed in the appendix (summarized in main text Fig. 2b).
>
> Fine-tuning doesn’t work?
>
> * On the contrary, we demonstrate that our methods works very well on fine-tuning in Figure 10 (appendix), e.g. +18% for ImageNet+Cars196→Cars196, and +7% for ImageNet+iFood→Food101. The performance is limited only in the *full-shot* fine-tuning scenario (Figure 11), although we still reliably outperform traditional distillation. The difference is how much data the downstream task has (which is outside the method’s control), not fine-tuning vs. fixed representation.
> * In general, pre-training and representation learning work best when the downstream task is data-deprived. When the downstream task has sufficient data, pre-training is not as useful. [a] This is a known limitation of our field, and many papers only focus on few-shot for downstream tasks. Note that Figure 10 tries to leverage knowledge learned from mid-sized datasets such as Cars196, and transfers downstream to a similarly mid-sized dataset split, so it is understandable that the improvement is small.
>     * [a] He, Kaiming, Ross Girshick, and Piotr Dollár. "Rethinking imagenet pre-training." ICCV 2019.
> * If reviewers consider it useful, we could swap the few-shot fine-tuning experiments into the main text and put full-shot fine-tuning into the appendix to better highlight this argument.
>
> Technical contribution is limited
>
> * We have other contributions:
>     * We propose a new task of trying to merge multiple representations into one without replay, and demonstrate performance similar to the with-replay oracle.
>     * We identify the issue that traditional distillation fails on downstream transfer.
> * Our technical contribution is a simple, surprising, easy to adopt once known, yet has been overlooked by the literature. It benefits anyone who want to learn a representation based on domain knowledge.

---

### Official Review · Reviewer_6iCZ · 2021-11-03

**Correctness:** 3
**Technical Novelty And Significance:** 2
**Empirical Novelty And Significance:** 2
**Recommendation:** 5
**Confidence:** 4

**Main Review:**

- In this paper, the authors have stated their key difference from the traditional distillation loss item is the generalist teacher item. Thus, I feel the contribution  is not that strong enough, since the generalist teacher is the pre-trained network.

- In addition, the generalist teacher item is just the model pretrained on imagenet dataset. What if in the other situation that the task is in NLP or speech recognition? Does it mean that the method can’t generalize well?

- In order to avoid catastrophic forgetting, do the authors try to add regularization item like the works of incremental learning do?

- The work here aims at transfer learning, but the experiments are set with the classes in the same domain.

- Some advice in the experiments, the chart figures are not very suitable here.


**Summary Of The Paper:**

- The authors propose a method of learning a consolidated image feature representation from a collection of related task-specific teachers that transfer well on novel recognition tasks.

- To achieve it, the authors utilize multi-teacher multi-task model distillation framework that jointly distills one or several task-specific teachers with a generalist one (trained with Imagenet dataset).

- Each teacher is set to operate on a different set of classes, and a multi-head student is trained to emulate all teachers.

- Experimental results show that the proposed method doesn’t need to revisit original training data of each teacher, but gains better performance.


**Summary Of The Review:**

- Besides the generalist teacher, what is the other contribution?

- Please discuss the generalization ability of the proposed method.

- Discuss more about the experiments.

---

> ### Author Response · Authors · 2021-11-19
> **Response to Reviewer 6iCZ**
>
> We thank the reviewer for the feedback and suggestions! We address the following concerns:
>
> Contribution
>
> * We respectfully disagree that a simple solution means a weak contribution. The *technical* part of our contribution is a simple, surprising, easy to adopt once known, yet has been overlooked by the literature (performs better than much more complicated, established methods like CRD on our goal). It benefits anyone who want to learn a representation based on domain knowledge.
> * In addition, we have other contributions:
>     * We propose a new task of trying to merge multiple representations into one without replay, and demonstrate performance similar to the with-replay oracle.
>     * We highlight that traditional distillation fails on downstream transfer and analyze its cause.
>
> Generalizing to non-vision tasks
>
> * If in an NLP task, the teacher networks are a well-trained model (e.g. BERT on Wikipedia) taken and fine-tuned on smaller domain data (e.g. medical text), we anticipate (but have not performed the experiment) that distilling the fine-tuned model can underperform the original well-trained model on a related but different (medical) dataset.
> * Obtaining a single model that performs well on vision and language (as in one single backbone that can take either images or text as input, rather than two separate backbones like CLIP) is beyond the scope of our paper.
> * Our work focuses on representation consolidation for visual recognition (which is already a rich domain).  We will happily re-work the writing to indicate that the paper’s scope is limited to vision classification tasks.
> * The paper aims to be a vision and classification-only work, and we will happily re-work the language to indicate that the paper’s scope is limited to vision classification tasks.
>
> Incremental learning methods
>
> * In a loose sense, our method can be seen as a form of the incremental learning method Learning without Forgetting applied to the original pre-training task (ImageNet). Other incremental learning methods may be able to replace our formulation, but is out of scope of our paper. We note that one of our contributions is identifying the problem of forgetting in representation distillation, and applying other incremental learning methods inherently builds on our finding.
>
> Are we transferring within the same-domain?
>
> * Our method transfers to other domains as well. One of our contributions is we transfer much better to unrelated domains compared to all distillation methods. Our method is on par with the generalist and the joint-training oracle — see non-bolded downstream datasets in Figure 2a and Figure 6.
> * Just to clarify: our work is on inductive transfer learning (e.g. pre-training + fine-tuning), not transductive transfer learning (i.e. domain adaptation). Domain adaptation tries to improve the performance of the *same* task on another domain, whereas our goal is to use knowledge in one task to help learning another task with a potentially different label space. In this kind of transfer learning, methods work better with related tasks (“similar domain”), but would still work on another domain with perhaps less efficacy.
>
> Appropriateness of figures
>
> * We would greatly appreciate any feedback that can improve the presentation in our paper. We tried to use bar plots with x-axis being different downstream tasks, but it was rather long and hard to interpret at a glance.
>
> “Discuss more about the experiments”
>
> * We will greatly appreciate any suggestions on adding discussions to our experiments! Is there any specific part that you would like us to write about?

---

### Decision · Program_Chairs · 2022-01-20

**Decision:**

Reject

**Comment:**

Authors present an approach to consolidate multiple teachers into a single student model that can be adapted to new tasks. The method involves using a proxy dataset to facilitate distillation to prevent having to replay images from the teacher datasets. A multi-task multi-head objective is utilized, agnostic to the loss function, in which two are studied. Downstream task performance is used as the performance measure.

Pros:
- The problem of how to best leverage multiple teachers for a downstream task is important and interesting.
- Presents a method to generate distilled students that can be finetuned to tasks that demonstrates performance gains over baselines (imagenet alone or task specific teacher).
- Easy to follow and implement.
- Analysis across multiple datasets.

Cons:
- Multiple reviewers expressed concerns about current level of novelty / contribution. In some sense, it is natural to expect that combinations of task-related and generalist distillation would improve performance.
- Main results demonstrate improvements in performance when teacher and tasks are related to one another. But authors do not address how to select task-specific teachers for distillation. Related tasks and their matching to the target task are assumed to be known. Authors cited related prior works that attempt to do this matching, but do not apply it to their study for a full solution.
- Authors do not study variations of generalist teachers. How does changing the generalist teacher impact performance?
- Some reviewers expressed concern presentation is not clear. In particular, the style of figures may not be appropriate to best convey results and analyses of this type of work. Comparing different approaches is difficult looking at thin lines. Tables are perhaps better suited to convey these results.
- Multiple reviewers expressed concerns full-finetuning results are not convincing (Fig 4), though few-shot results look more convincing

Authors and reviewers had interaction, but reviewers maintained their recommendation of weak reject. All reviews unanimous in their decisions. Authors are encouraged to take into consideration all the comments and submit to another venue.